# Diversity patterns in terrestrial tetrapod clades are governed by equilibrium dynamics

Felipe O. Cerezer[1,2]*, Antonin Machac[2], Jan Smyčka[1,3], Iñigo Rubio-López[2,4], Maxime Quétin[2,4], David Storch[1,4]*

1 Center for Theoretical Study, Charles University, Prague, Czech Republic, 2 Laboratory of Environmental Microbiology, Institute of Microbiology of the Czech Academy of Sciences, Prague, Czech Republic, 3 Department of Biological Sciences, Simon Fraser University, Burnaby, Canada, 4 Department of Ecology, Faculty of Science, Charles University, Prague, Czech Republic

* cerezerfelipe@gmail.com (FOC); storch@cts.cuni.cz (DS)

## Abstract

Large-scale patterns of species richness have been attributed to ecological limits, variation in diversification rates, and differences in evolutionary time, yet the relative importance of these drivers remains debated. Here, we present a unifying framework distinguishing four richness-generating scenarios, defined by contrasting roles of evolutionary time and speciation rates, which yields explicit and testable predictions for how evolutionary time, speciation, and environmental factors influence species richness. We applied this framework by analyzing 129 distinct, nonoverlapping clades spanning amphibians, reptiles, birds, and mammals. For each clade, we integrated historical biogeographic reconstructions, multiple estimates of speciation rates, and GIS-based environmental data. Using structural equation modeling, we quantified the direct and indirect effects of evolutionary time, speciation rates, and environmental conditions (productivity, temperature, and precipitation) on species richness. We further tested whether these effects varied systematically with clade-level traits, including age, physiology, diversity, and geographic extent. Productivity emerged as the dominant predictor of species richness, exerting strong and consistent direct effects that were largely invariant to clade traits. In contrast, speciation rates contributed little to species richness, while the influence of evolutionary time was highly context-dependent and most pronounced in younger clades. Temperature showed consistent direct effects not mediated by productivity, evolutionary time, or speciation rates, whereas precipitation influenced richness primarily via productivity. Together, our results support a productivity-driven equilibrium view of species richness, in which diversity reflects a balance between speciation and extinction regulated by energy availability. Deviations from equilibrium dynamics, particularly in younger clades, highlight the role of evolutionary history on biodiversity gradients.

**Data availability statement:** All numerical data underlying Figs 4–7A, and Figs A–AW in S1 Appendix, and the *R* scripts used to generate the figures, are publicly available in the Zenodo repository under the Attribution 4.0 International license (DOI: https://doi.org/10.5281/zenodo.14008084). The repository includes CSV and RDS files containing the data used in the analyses and figures, along with *R* scripts that reproduce the figures presented in the manuscript. Some of the data compiled in the repository, such as phylogenies, species distribution data, and environmental layers, were obtained from previously published sources and are fully cited in the Methods section.

**Funding:** FOC and DS were supported by the Czech Science Foundation (GAČR, 20-29554X; https://gacr.cz). FOC, AM, IRL, and MQ were supported by the Czech Science Foundation (GAČR, 23-05977S; https://gacr.cz). The funder had no role in study design, data collection and analysis, decision to publish, or preparation of the manuscript.

**Competing interests:** The authors have declared that no competing interests exist.

## Introduction

Understanding why some regions harbor more species than others remains a central question in ecology and evolution [1,2]. Classic explanations for large-scale species richness patterns typically fall into three broad categories: (1) ecological limits, where the environment sets a carrying capacity for species coexistence [3–5]; (2) diversification rates, reflecting spatial variation in how quickly species originate and go extinct [1,6]; and (3) evolutionary time, namely the time lineages have had to diversify in a region [7,8]. However, both classical syntheses [9–11] and more recent work [2,12–16] have shown that these explanations are not mutually exclusive, but instead reflect interacting processes that jointly influence species richness. This recognition motivates the need to move beyond the traditional trichotomy toward integrative frameworks that explicitly consider their interplay.

Empirical tests of processes proposed to affect large-scale diversity variation are often fragmented, frequently isolating individual mechanisms or focusing on single clades. A central debate concerns whether large-scale richness patterns are driven mainly by evolutionary time or diversification: some studies emphasize evolutionary time as a primary predictor of richness [17–20], whereas others emphasize variation in diversification rates [21,22]. However, support for diversification-based explanations has proven highly inconsistent, with evidence suggesting strong context dependence across clades and regions [23–27]. At the same time, a large body of work documents strong correlations between species richness and environmental factors such as energy availability and productivity [3,28–31]. Yet these environmental relationships are frequently evaluated independently of evolutionary processes, despite growing evidence that ecological limits can modulate both diversification and the time available for speciation [13,32,33]. Together, these findings suggest that no single mechanism provides a sufficient explanation, highlighting the need for integrative models that simultaneously evaluate multiple pathways linking environment, evolutionary history, and diversification to species richness.

Although an interactive view of these processes is increasingly accepted, empirical implementations differ substantially in how these interactions are conceptualized and tested. For example, ecological limits are often viewed as hard ceilings imposed by finite niches or resources [34], yet they can also emerge from diversity-dependent diversification, with higher speciation or lower extinction rates potentially shifting these limits upward [1,35,36]. Similarly, the timing of colonization can interact with environmental conditions: regions colonized earlier may benefit from prolonged climatic stability or sustained productivity, promoting extended diversification [16,37]. Despite being central to theoretical explanations of large-scale species richness patterns, these interactions are rarely evaluated simultaneously, and most empirical studies examine only subsets of mechanisms in isolation. This fragmentation limits our ability to determine which processes dominate in their effects on species richness patterns and under what conditions, highlighting the need for approaches that can jointly assess historical, evolutionary, and environmental drivers of species richness [2,15,36].

To provide such an integrative perspective, we propose a conceptual scheme that classifies species richness-generating hypotheses along two orthogonal axes (Fig 1).

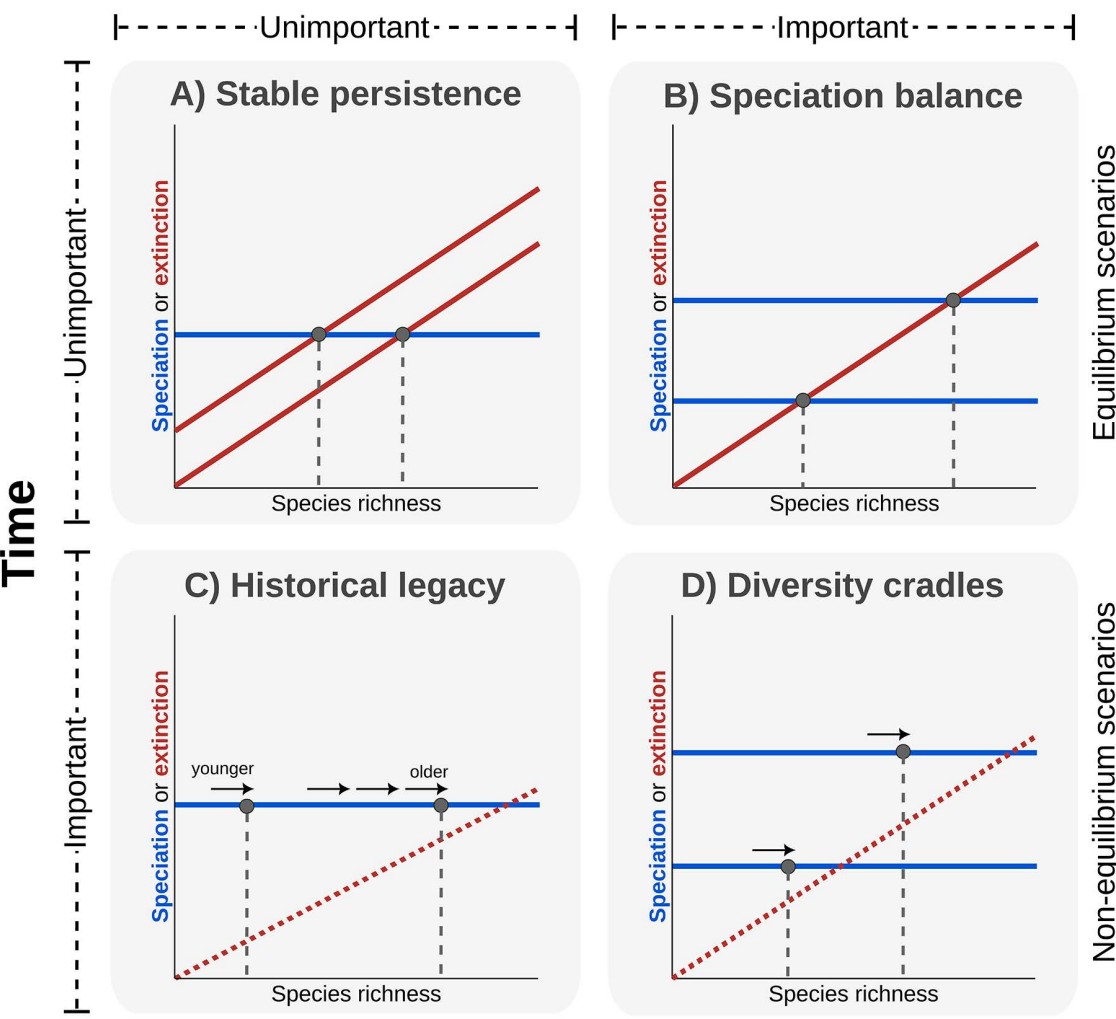

**Fig 1. Conceptual framework distinguishing four scenarios for the generation of large-scale species richness patterns.** The framework adopts a dynamical perspective that assumes some biodiversity equilibria may exist but can be irrelevant in certain scenarios. Scenarios are defined by two criteria: (1) whether species richness reflects equilibrium dynamics or is primarily shaped by historical, non-equilibrium processes (time for species accumulation), and (2) whether spatial variation in richness is primarily driven by differences in speciation rates. Together, these criteria define four scenarios. **A)** *Stable persistence*, in which regional richness is determined by current environmental conditions driving differences in extinction propensities, with little influence of speciation rates or evolutionary time. **B)** *Speciation balance*, where differences in equilibrium richness arise primarily from variation in speciation rates. **C)** *Historical legacy*, in which species richness is out of equilibrium and reflects the time available for species accumulation. **D)** *Cradles of diversity*, where richness is determined by speciation rates without reaching equilibrium, depending on both speciation rates and available time. The lines in the panels show the relationships between species richness and per-species speciation (blue) and extinction (red) rates. For simplicity, extinction is shown as the only diversity-dependent process (speciation rates are therefore depicted as horizontal lines). The extinction curve is irrelevant for the two nonequilibrium scenarios (C–D), where it is depicted by dashed lines. Although the framework includes both speciation and extinction, empirical analyses focus on speciation rates because extinction cannot be reliably estimated from extant-only phylogenies. Nevertheless, theoretical expectations about extinction dynamics allow us to interpret broad-scale patterns in richness and evaluate scenario predictions.

The first axis distinguishes equilibrium scenarios, where species richness reflects a long-term dynamic balance between speciation and extinction, from nonequilibrium scenarios, where richness reflects a combination of historical processes (e.g., colonization events) and lineage properties, such as clade age [5,38]. The second axis separates mechanisms in

which richness is directly linked to spatial variation in speciation rates from those where richness varies independently of speciation [1,6,39]. Although multiple processes may act in concert in governing species richness, and additional dimensions could further refine this framework, combining these two axes yields four broad, conceptually distinct richness-generating scenarios. These scenarios emphasize contrasting roles of evolutionary time and speciation rates, while simultaneously generating explicit, testable predictions for how environmental factors are expected to influence species richness (Figs 1 and 2).

In the *Stable persistence* scenario, species richness remains near equilibrium, unaffected by colonization time or variation in speciation rates (Fig 1). Richness primarily reflects environmental constraints on coexistence, particularly productivity, and energy availability [3,5,28], with evolutionary time and speciation rates playing minimal roles (Fig 2). By contrast, the *Speciation balance* scenario also assumes equilibrium but attributes richness differences to variation in speciation rates (Fig 1). Colonization time has little effect, and environmental factors influence richness indirectly by affecting speciation rates [36] (Fig 2). The *Historical legacy* scenario proposes that richness is mainly determined by assemblage age or colonization history, with limited influence of speciation rates (Fig 1). Under this view, older clades or regions colonized earlier tend to be more species-rich [7,8,40], while productivity has weak or negligible direct effects (Fig 2). Instead, environmental factors such as temperature and precipitation affect richness indirectly, as species can preferentially colonize warmer and more humid regions due to the lineage's tropical origin and niche conservatism [37]. Lastly, the *Cradles of diversity* scenario describes a nonequilibrium state in which richness differences among regions are primarily driven by variation in speciation rates, without reaching any equilibrium. Colonization time plays a secondary role, and environmental effects on richness are mediated through speciation rates (Fig 2). This scenario conceptually overlaps with the 'cradle' aspect of the Out-of-the-Tropics hypothesis, which identifies the tropics as regions where many new species originate [7]. Unlike the traditional trichotomy [2], this classification integrates historical, diversification, and environmental processes into a unified set of testable hypotheses (Figs 1 and 2).

Evaluating how support for the four richness-generating scenarios varies across biological systems requires analyses conducted at the level of well-defined clades. Clades represent natural evolutionary units within which diversification history and ecological constraints operate jointly, making them particularly suitable for linking macroevolutionary and macroecological processes to observed patterns of species richness [41–44]. Importantly, clades differ systematically in physiological, evolutionary, and geographic attributes, and these differences are expected to shape both the mechanisms generating richness and the relative support for alternative scenarios. For example, thermal physiology is expected to modulate the strength of temperature–richness relationships. Ectothermic clades, which depend directly on environmental temperature to regulate metabolism, are predicted to show stronger temperature–richness associations, whereas endothermic clades may be more strongly limited by resource availability, such as primary productivity [45–47]. Evolutionary history, reflected by clade age and diversity, captures both the time available for diversification and the extent to which that potential has been realized [48,49]. Younger, species-poor clades may still be undergoing expansion, whereas older, species-rich clades are more likely to have experienced diversification slowdowns or reached ecological limits [5,19,38,50,51]. Finally, geographic extent captures the spatial arena over which diversification unfolds, with broader ranges exposing clades to greater environmental variation that can influence richness patterns [11,52,53]. By explicitly linking support for alternative richness-generating scenarios to clade-level traits, we test whether equilibrium and nonequilibrium dynamics differ systematically among evolutionary lineages.

Here, we evaluate support for the four richness-generating scenarios across amphibians, reptiles, birds, and mammals. To avoid arbitrary clade selection [20,54], we employ a topology-based approach to delimit clades from large-scale phylogenies. This approach identifies clades based on homogeneity in diversification structure, allowing replication across multiple independent clades while minimizing biases associated with arbitrary or nested clades. We then use structural equation models (SEMs) to estimate the direct and indirect effects of evolutionary time, speciation rates, and environmental variables (temperature, precipitation, and productivity) on species richness. To capture historical constraints on regional

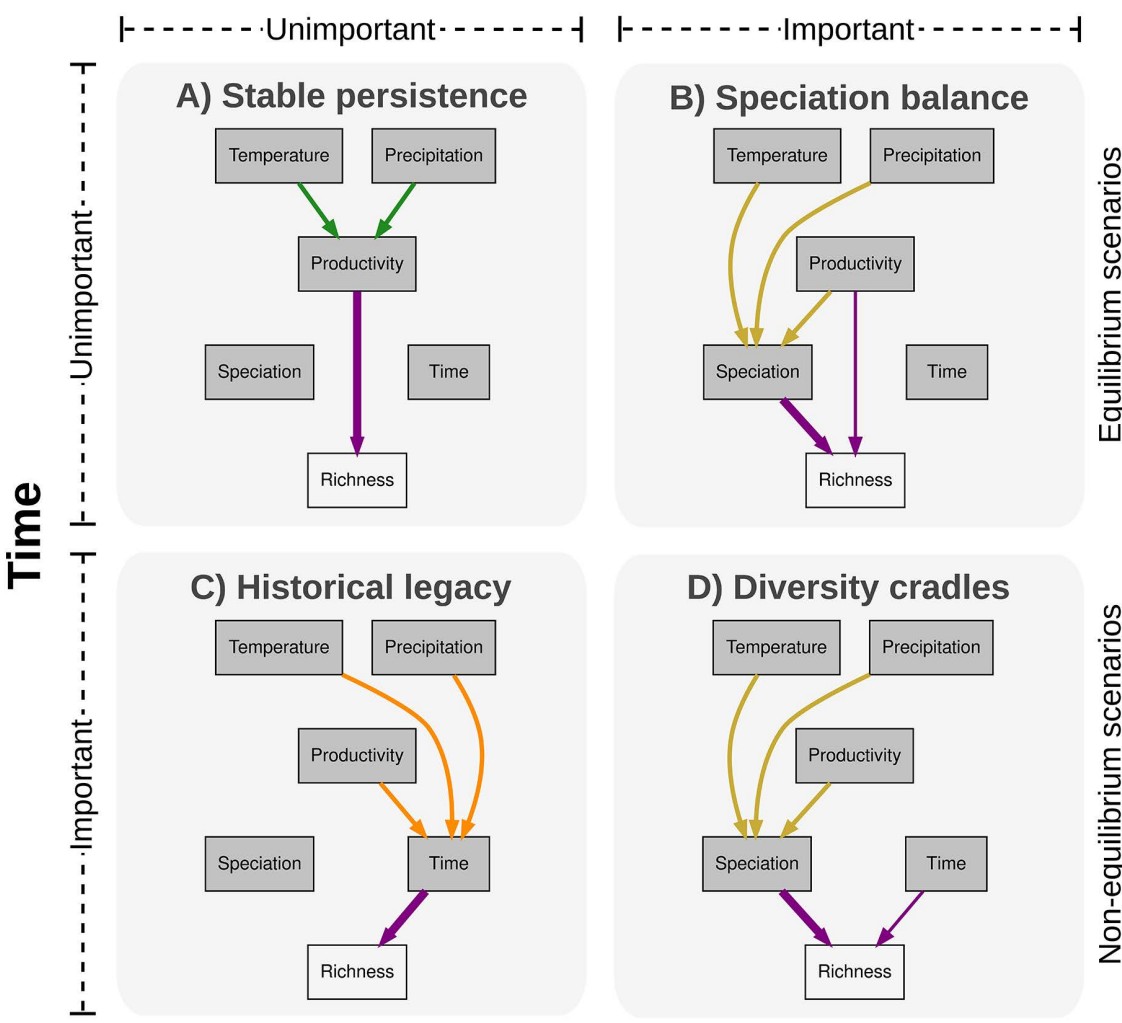

**Fig 2. Diagnostic predictions of four species richness-generating scenarios.** For the *Stable persistence* scenario: (1) speciation rate is not related to species richness, (2) evolutionary time has negligible effect, (3) productivity has a strong direct effect on richness, (4) direct effects of climate are weak, and (5) climate affects richness indirectly via productivity. In the *Speciation balance* scenario: (6) speciation rate is positively related to richness, (7) evolutionary time has little effect on richness, (8) climate (especially temperature) affects richness primarily via speciation rate, (9) productivity influences richness mainly indirectly via speciation, and (10) indirect effects of temperature via speciation are stronger in ectotherms*. Under the *Historical legacy* scenario: (11) speciation rate is not related to richness, (12) evolutionary time has a strong positive effect on richness, and (13) climate and productivity can affect richness indirectly by modulating the time available for colonization and species accumulation. In the *Cradles of diversity* scenario: (14) speciation rate is positively related to richness, (15) climate and productivity primarily influence richness indirectly via speciation rates, (16) richness is determined by the product of age and speciation rate, and (17) ectotherms show stronger temperature-mediated effects on speciation*. Because some predictions are not exclusive to a single scenario, their evaluation should rely on the simultaneous testing of multiple predictions. The pathways highlighted for each scenario correspond to its idealized ("pure") formulation, representing the pathways expected to dominate under that scenario. Additional pathways may also operate but are predicted to be comparatively weaker. Predictions marked with an asterisk (*) apply under the assumptions of the Metabolic Theory of Ecology. More details on these scenarios are shown in Fig 1.

richness, we integrate ancestral range reconstructions to estimate clade arrival times across regions. Finally, we assess whether support for alternative scenarios varies systematically with clade traits, including thermal physiology, clade age, clade diversity, and geographic extent. By combining conceptual predictions, replicated clade-level analyses, and

comparisons across clade-level traits, our study provides a unified framework for evaluating how history, diversification, and environment jointly shape global patterns of species richness.

## Methods

Our analyses aim to evaluate how historical (assemblage age), diversification (speciation rates), and environmental (temperature, precipitation, and net primary productivity) factors shape species richness across terrestrial tetrapod clades. We first delineated clades using the Laplacian spectra of large-scale phylogenies, a graph-theoretic approach that identifies nonoverlapping sets of species based on similarities in branching structure [55]. For each clade, we mapped species richness across equal-area grid cells and quantified historical, diversification-related, and environmental predictors at this spatial resolution. Historical influences were captured as the timing and likely regional arrival of lineages into assemblages, reconstructed using historical biogeographic models, whereas diversification was represented by speciation rates. Environmental conditions were characterized by temperature, precipitation, and net primary productivity as proxies for climate and energy availability. For each clade, we applied SEMs to estimate the direct effects of historical, diversification-related, and environmental variables on species richness, as well as three key indirect pathways: (1) environmental effects on species richness mediated through variation in speciation rates, reflecting, for instance, temperature-dependence of metabolic rates and population dynamics [28,45,56]; (2) environmental effects on species richness mediated through evolutionary time (lineage arrival times), reflecting how environmental conditions facilitated or constrained successful colonization and establishment of lineages [16,33,37]; and (3) climatic effects on species richness mediated through productivity, reflecting the role of climate in regulating energy availability [57–59]. Finally, we tested whether the strength of these pathways varied systematically with clade-level traits, including age, thermal physiology, diversity, and geographic extent.

### Species distribution data and phylogenies

Geographical distributions of terrestrial tetrapods were obtained from available databases: birds from [60], mammals and amphibians from [61], and reptiles from [62]. Species range maps were aggregated into equal-area grid cells at a resolution approximating 1° × 1° (≈110 km at the equator), a spatial grain that is widely used in large-scale macroecological analyses of vertebrate diversity and facilitates standardized comparisons across taxa and regions [24,63–67]. Species richness for each cell was calculated by summing the number of species whose range maps overlapped that cell (Fig A in S1 Appendix). Spatial handling procedures were performed using the R packages *'letsR'* [68], *'rgdal'* [69], and *'sf'* [70].

The phylogeny of birds was derived from [24], mammals from [71], and amphibians from [72]. The squamates' phylogeny was based on [73], while the phylogeny of turtles and crocodilians followed [74]. Using tools provided by the VertLife project (https://vertlife.org/), we randomly sampled a set of posterior distributions of trees ($n = 2,000$) and computed the maximum-clade credibility (MCC) topology for each tetrapod group. We then pruned the phylogenies to include only species with distribution data, resulting in a sample size of nearly 30,000 tetrapod species (9,234 birds, 5,145 mammals, 6,361 amphibians, and 9,129 reptiles) (Table A in S1 Appendix). The MCC tree was computed using the *'phangorn'* package [75], and species pruning was done using the *'phytools'* package [76].

### Clade selection

Because the drivers of species richness can vary among independent evolutionary lineages, we focused our analyses on clades delineated using a formal algorithm, rather than treating each tetrapod class as a single unit. This design allows direct comparison of richness-generating processes across multiple, independent evolutionary lineages. Numerous approaches have been proposed to delineate clades for macroecological analyses [54,77], including the use of taxonomic ranks [19], temporal slicing of phylogenies [78], and random node sampling [20]. However, each approach has limitations. Taxonomic ranks are inherently arbitrary and often constrain clades to similar ages, reducing variation in evolutionary time

and potentially biasing analyses that include time as a predictor [20]. Random node sampling may select nested clades, causing species to appear in multiple analytical units and violating assumptions of statistical independence [77].

To overcome these limitations, we delineated clades using the Laplacian spectrum approach [55], which does not rely on taxonomic ranks or predefined temporal thresholds. This method applies graph-theoretic principles to phylogenetic trees, using spectral density profiles to summarize global branching structure and identify sets of species that share similar diversification dynamics [55]. By clustering these profiles, the approach identifies nonoverlapping clades that are homogeneous in their branching patterns and diversification dynamics (Fig 3).

For each tetrapod class, we initially delineated 50 spectral modalities using $k$-means clustering [79] of Laplacian spectral profiles, as implemented in the 'RPANDA' package [80]. From these initial modalities, we retained only clades containing at least 50 species. This minimum size threshold was applied to ensure stable estimation of relationships between species richness and its potential drivers, as very small clades tend to exhibit limited variation in richness and can yield unreliable effect estimates. Although both the number of initial modalities and the minimum clade size are necessarily arbitrary, this combination represents a compromise between statistical robustness, replication across independent clades, and broad taxonomic coverage. Importantly, this filtering step excluded only a small fraction of species from the matched phylogeny–distribution datasets (≤2.2% across tetrapod classes), suggesting that the criterion is unlikely to bias results against recently diversified or species-poor lineages.

The Laplacian spectrum approach can yield both monophyletic and paraphyletic clades, as it prioritizes similarity in branching dynamics across the entire tree [55]. In this framework, clades are defined by homogeneity in diversification structure, not by shared taxonomy, traits, or geographic distributions. Consequently, when a subset of lineages within a traditionally defined clade exhibits distinct diversification dynamics, it is preferable to treat it separately rather than include it within a larger clade whose richness-generating processes may differ. Paraphyletic clades are therefore not conceptually different from monophyletic clades that necessarily exclude extinct or unsampled sublineages [81].

To evaluate the sensitivity of our analyses to methodological choices in clade delineation, we repeated the entire analytical pipeline using alternative numbers of initial spectral modalities (40 and 60) and conducted additional analyses excluding paraphyletic clades. In all cases, clades were redefined and all downstream analyses were rerun using the same procedures as in the main analyses. We further evaluated the robustness of our results to phylogenetic uncertainty by repeating all analyses using phylogenies constructed exclusively from molecular data, thereby excluding species whose placements were inferred from taxonomy. For each tetrapod class, we used molecular-only MCC phylogenies (birds: [24]; mammals: [71]; amphibians: [72]; squamates: [73]; turtles and crocodilians: [74]), pruned to retain only species with available distribution data. Using these molecular-only phylogenies, we reapplied the Laplacian spectrum approach and reran all SEMs following the same analytical framework as in the primary analyses.

## Environmental variables

We quantified environmental conditions within grid cells using mean annual temperature, mean annual precipitation, and net primary productivity (NPP) (Fig B in S1 Appendix). These variables were selected because they capture complementary dimensions of climate and energy availability that underpin the richness-generating scenarios evaluated in this study (Fig 2). Temperature and precipitation influence physiological tolerances, metabolic rates, and geographic constraints on species distributions [28,30,47], with temperature also playing a potential role in determining speciation rates [45]. NPP represents a measure of biologically available energy and resource supply, which has long been linked to species richness and coexistence dynamics [28,47,52,82]. Temperature and precipitation data were obtained from WorldClim long-term averages (1970–2000) at 2.5 arc-minute resolution [83]. NPP data were derived from the MODIS/Terra Net Primary Production Gap-Filled Yearly L4 product (MOD17A3HGF, Version 6.1) at 500 m resolution, averaged across years 2001–2024 [84]. This product applies quality controls and gap-filling procedures to ensure spatial and temporal consistency. Raster processing and extraction of environmental values to grid cells were conducted in R using the package 'terra' [85].

**Step 1. Clade delineation**     **Step 2: Mapping species richness of a hypothetical clade**

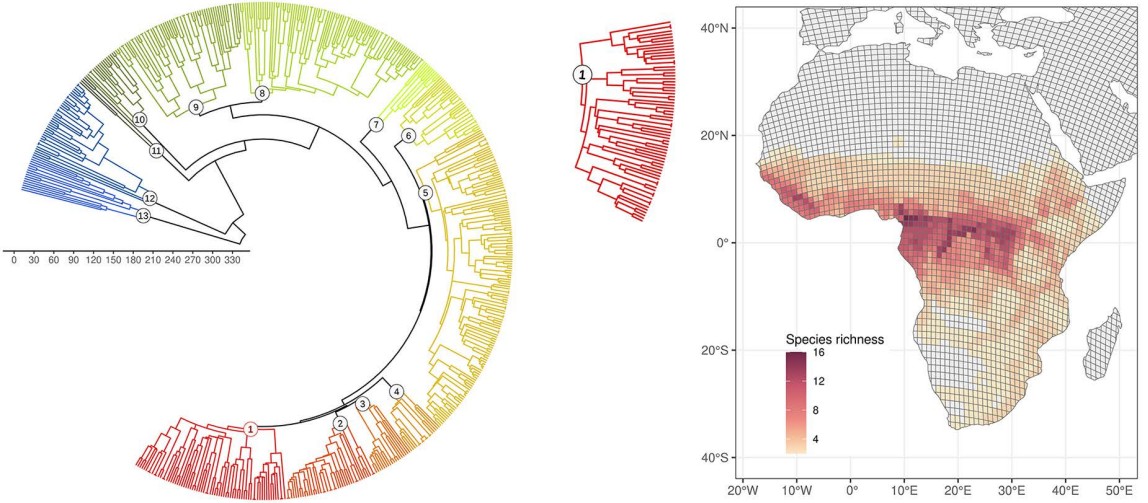

**Step 3: Structural equation modeling of richness drivers**

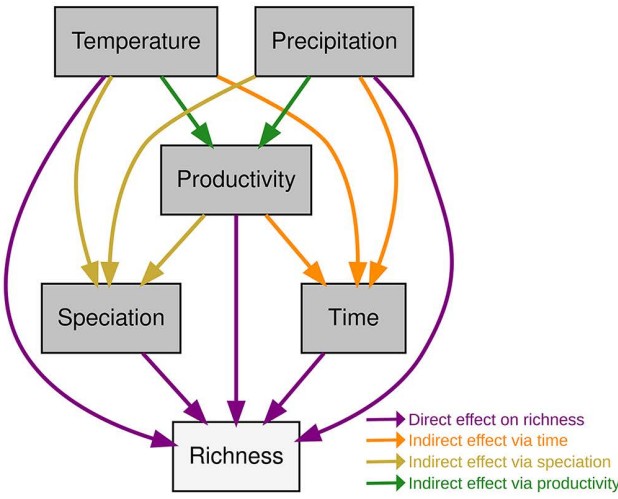

**Fig 3. Schematic overview of the analytical workflow used to test richness-generating scenarios across tetrapod clades.** All elements are illustrative. Step 1. Clade delineation using Laplacian spectral decomposition: clades were delineated separately for each tetrapod class (amphibians, reptiles, birds, and mammals) using a data-driven approach that identifies nonoverlapping evolutionary units from phylogenies. Step 2. Mapping species richness of a hypothetical clade (Clade 1): species belonging to the clade are mapped to quantify spatial variation in richness, which serves as the response variable in subsequent analyses. Step 3. Structural equation modeling of richness drivers: species richness was tested against evolutionary time (assemblage age), speciation rates, and environmental predictors (temperature, precipitation, and net primary productivity) using structural equation models (SEMs) that estimate both direct and indirect effects. Predictor variables were compiled at the same spatial resolution as richness (see Methods), and SEMs were fitted independently for each clade. Arrow colors indicate different pathways: direct effects on species richness (purple), indirect effects mediated by evolutionary time (orange), speciation rates (golden yellow), and productivity (green). The geographic panel uses continent boundaries from Natural Earth Admin 0 Country Boundaries (https://www.naturalearthdata.com), which is in the public domain (CC0) and compatible with the CC BY 4.0 license. The map was generated in R using this base layer.

## Estimating evolutionary time

To quantify the evolutionary time available for species accumulation in each assemblage (i.e., grid cell), we applied a model-based metric of community age that integrates historical biogeography and ancestral range reconstruction (Fig

C in S1 Appendix) [86–88]. This approach estimates the time at which lineages first colonized a given region, providing a process-informed measure of assemblage age [86]. Traditional phylogenetic metrics, such as mean pairwise distance (MPD) or mean branch length (MBL), have frequently been used as proxies for evolutionary time [12,14,18,27,89], but they are strongly influenced by diversification dynamics and dispersal–extinction processes, making assemblage age and diversification rate intrinsically inseparable [90]. For comparison, we also explored MPD and MBL as alternative predictors of evolutionary time in sensitivity analyses (see Results section).

For each clade, we first defined biogeographic regions based on phylogenetic turnover, using the calc_evoregion function in the 'Herodotools' package [86]. This method identifies regions of independent lineage diversification (evoregions) by integrating a phylogenetic fuzzy weighting matrix with discriminant analysis of principal components and *k*-means clustering [91]. To maintain computational tractability, we limited the number of evoregions to a maximum of 10 per clade. While biogeographic realms or climatic zones (e.g., biomes) are often used to define regions, such predefined units may not align with the evolutionary history or diversification patterns of specific clades [91]. In contrast, delineating regions based on phylogenetic turnover allows us to identify areas where closely related species co-occur, capturing lineage-specific biogeographic structure and evolutionary distinctiveness. This clade-specific approach is particularly important when comparing clades with differing histories of dispersal and diversification across space.

We then reconstructed ancestral ranges using the Dispersal–Extinction–Cladogenesis (DEC) model implemented in BioGeoBEARS [87,92]. The DEC model is a likelihood-based framework that estimates lineage range dynamics over time by modeling dispersal (range expansion), local extinction (range contraction), and cladogenesis (speciation accompanied by range subdivision) [87]. Although alternative models exist (e.g., DEC+J, BAYAREALIKE, and DIVALIKE), we chose DEC for its interpretability and ability to capture key biogeographic processes without overparameterization [93]. Using these reconstructions, we calculated assemblage age with the calc_age_arrival function in the 'Herodotools' R package [86], following the method described in [88]. For each species in a grid cell, we traced its phylogeny to identify the deepest ancestral node whose reconstructed range included the focal region. The age of this node was considered the species' arrival time in the assemblage. When no such node was identified, implying that the species dispersed into the region after its most recent speciation, we assigned a minimal arrival time of 10 years, indicating very recent colonization (see [88] for more details). Assemblage age was computed as the mean arrival time across all species in a cell.

Ancestral range reconstructions cannot recover exact historical distributions, as they infer past ranges from present-day occurrences conditioned on a time-calibrated phylogeny and model assumptions [87,92]. Accordingly, our objective is not to reconstruct absolute historical locations or precise timing of past events, but to estimate the relative timing of lineage colonization across regions. Assemblage age therefore represents a model-based approximation of when lineages likely first occupied a region, interpreted in a comparative rather than absolute sense. This formulation allows us to test whether species-rich regions are associated with longer lineage residence times relative to other regions, which is relevant for evaluating evolutionary time hypothesis.

## Estimating speciation rates

To assess speciation rates, we used three complementary approaches: the DR statistic, Bayesian Analyses of Macroevolutionary Mixtures (BAMM), and Cladogenetic Diversification Rate Shift model (ClaDS). The DR statistic, derived from the inverse of the equal splits measure, offers a species-specific assessment by considering splitting events and internode distances along the root-to-tip path [24]. Although originally proposed as an estimator of net diversification, subsequent work has demonstrated that the DR statistic more closely reflects recent speciation rates [94]. BAMM enables the modeling of complex speciation and extinction dynamics by identifying heterogeneous shifts in evolutionary rates across a time-calibrated phylogeny [95]. BAMM was run with four MCMC chains, each lasting 10 million generations and sampling every 2000 generations. We accounted for incomplete taxon sampling and discarded the first 25% of samples as burn-in. Speciation-extinction priors (Table B in S1 Appendix) and tip-based speciation rates were obtained using the 'BAMMtools'

package [96]. ClaDS is a flexible Bayesian framework that estimates lineage-specific speciation rates by assuming that diversification changes occur gradually and heritably along branches, thereby capturing fine-scale rate variation without invoking discrete rate shifts [97]. We applied the Julia implementation of ClaDS, which uses data augmentation to accelerate computation and improve inference of diversification dynamics [98].

We focus on tip-based speciation rates because diversification rates inferred from extant-only timetrees are mathematically nonidentifiable through deep time [99]. In particular, extinction rates and net diversification trajectories cannot be reliably estimated without additional data or strong biological constraints [99], which fundamentally limits inference on historical diversification dynamics from molecular phylogenies alone. In contrast, recent (tip) speciation rates are comparatively robust, whereby studies show that recent speciation dynamics can be accurately recovered even when deeper-time rate estimates are unreliable [94,100,101]. While we acknowledge that current speciation rates may not fully reflect the clade's entire diversification history, tip-based estimates derived from DR, BAMM, and ClaDS represent the most conservative speciation measures available given the inherent nonidentifiability of past rates.

To explore spatial patterns, we calculated the median of these tip rates for co-occurring species within each grid cell. DR, BAMM, and ClaDS tip rates were all computed for comparison, and DR was adopted as the primary metric for downstream analyses due to its computational efficiency, flexibility, and widespread validation as a proxy for recent speciation [94,102]. Extinction rates were not directly estimated due to their unreliability in extant-only timetrees [99,103]. However, we considered the conceptual implications of alternative extinction regimes, such as differences in extinction propensity among regions, with higher extinction propensity in species-poor, low-energy temperate regions (sensu [90]) (Fig 1).

## SEMs

To evaluate support for the four richness-generating scenarios (*Stable persistence*, *Speciation balance*, *Historical legacy*, and *Cradles of diversity*), we used SEMs to quantify the direct and indirect effects of historical, diversification, and environmental variables on species richness (Fig 3). For each clade, we fitted SEMs that jointly estimated the effects of temperature, precipitation, and productivity, together with evolutionary time (assemblage age) and diversification (speciation rates). The model structure was designed to capture all biologically plausible causal pathways linking evolutionary history, diversification dynamics, climate, and energy availability to spatial variation in species richness, allowing multiple mechanisms to be evaluated simultaneously within a single comparative framework.

In addition to direct effects on species richness, the SEMs included three classes of indirect pathways (Fig 3). Climate variables (temperature and precipitation) were allowed to affect richness through speciation rates, evolutionary time (assemblage age), or productivity, while productivity was also allowed to affect richness indirectly via speciation rates or evolutionary time. These paths represent hypothesized links among lineage persistence, diversification, climate, and energy availability that have been widely proposed in macroecology [2,28,37,45,56]. Indirect effects were quantified by multiplying standardized path coefficients along each causal chain. Species richness was natural-log transformed to improve normality and homoscedasticity, and all predictor variables were z-transformed to facilitate comparison of effect sizes across paths. Multicollinearity among predictors was evaluated using variance inflation factors (VIF). Across all clades, VIF values remained below the commonly used threshold of 5, indicating that collinearity was unlikely to bias parameter estimates (Figs D–G in S1 Appendix).

SEMs were fitted separately for each clade using the '*lavaan*' package in R [104]. Model performance was evaluated using standard goodness-of-fit indices, including the Comparative Fit Index (CFI), Tucker–Lewis Index (TLI), Root Mean Square Error of Approximation (RMSEA), and Standardized Root Mean Square Residual (SRMR) [105,106]. While these indices provide useful diagnostics, our primary emphasis was on estimating theoretically motivated and biologically interpretable pathways that directly correspond to alternative richness-generating mechanisms, rather than on optimizing

absolute model fit. In addition to the full integrative SEMs, we assessed relative support for the four richness-generating scenarios using a model-selection approach. For each clade, we fitted four reduced, scenario-specific SEMs corresponding to the *Stable persistence*, *Speciation balance*, *Historical legacy*, and *Cradles of diversity* hypotheses, retaining only those pathways predicted to dominate under each scenario (Figs 1 and 2). We then compared these competing models using both Akaike Information Criterion (AIC) and Bayesian Information Criterion (BIC). For each clade, the scenario associated with the lowest information criterion value was interpreted as receiving the strongest empirical support. This analysis allowed us to quantify how frequently each scenario best explains richness patterns across tetrapod clades, directly linking the conceptual framework to the empirical results.

Since global biodiversity patterns are well known to exhibit strong spatial autocorrelation [107,108], we evaluated whether spatial structure influenced our results. Standard path-model implementations do not provide a straightforward or widely accepted approach for incorporating spatial covariance in multi-response models fitted repeatedly across many clades. Attempts to incorporate spatial covariance structure in such contexts can lead to convergence failures or biologically uninterpretable parameter estimates [107,109]. We therefore assessed spatial autocorrelation by calculating Moran's I on the residuals of each clade-specific model and, as expected, detected strong spatial structure (Fig H in S1 Appendix). To pragmatically assess the sensitivity of our inferences to broad-scale spatial structure, we re-fitted all SEMs with latitude and longitude included as additional predictors.

### Influence of clade traits on richness-generating processes

We further tested whether support for different richness-generating scenarios varied systematically with clade-level traits. Specifically, we considered clade thermal physiology, age, diversity, and geographic extent. Thermal physiology was classified into endothermic and ectothermic clades. Clade age was defined as the crown age of each clade, clade diversity as the total number of species, and geographic extent as the number of grid cells occupied by each clade.

To quantify how these clade-level traits modulate richness-generating processes, we fitted separate linear regressions relating each trait to the estimated strengths of the direct effects of evolutionary time, speciation rates, and environmental predictors obtained from the SEMs. We focused on emergent properties at the clade-level, treating non-overlapping clades as independent evolutionary lineages. This approach enables robust cross-tetrapod comparisons while avoiding potential methodological artifacts that can arise when applying phylogenetic regressions to merged trees differing in calibration, inference framework, or branch length scaling [110,111]. By analyzing discrete clades of varying taxonomic rank and evolutionary depth, our framework captures large-scale patterns of species richness in a manner widely adopted in macroecology and macroevolution [42–44,48].

## Results

Our analyses included 129 nonnested tetrapod clades with more than 50 species, delineated using 50 Laplacian spectral modalities: 34 amphibian clades, 34 reptile clades, 31 bird clades, and 30 mammal clades (Figs I–L in S1 Appendix). Retained clades contained on average 227 species (range: 57–728; Fig M in S1 Appendix), had a mean crown age of 65 million years (range: 10–266 Ma; Fig M in S1 Appendix), and occupied geographic areas ranging from $7 \times 10^5$ to $1.3 \times 10^8$ km² (Fig N in S1 Appendix).

Analyses across these clades revealed that productivity (NPP) had the strongest and most consistent direct effect on species richness (Figs 4; O in S1 Appendix). Evolutionary time and temperature also influenced species richness, but their effects were generally weaker, more variable, and clade-specific (Figs 4; O in S1 Appendix). In contrast, speciation rates explained little variation in richness and were in some cases weakly negative (Figs 4; O in S1 Appendix). Indirect effects were generally minor (Figs 4; P–R in S1 Appendix), with the only consistently detectable pathway being a positive influence of precipitation on richness mediated via productivity (Figs 4; P in S1 Appendix). Other indirect effects via speciation rates or evolutionary time were negligible (Figs 4; Q–R in S1 Appendix).

PLOS Biology

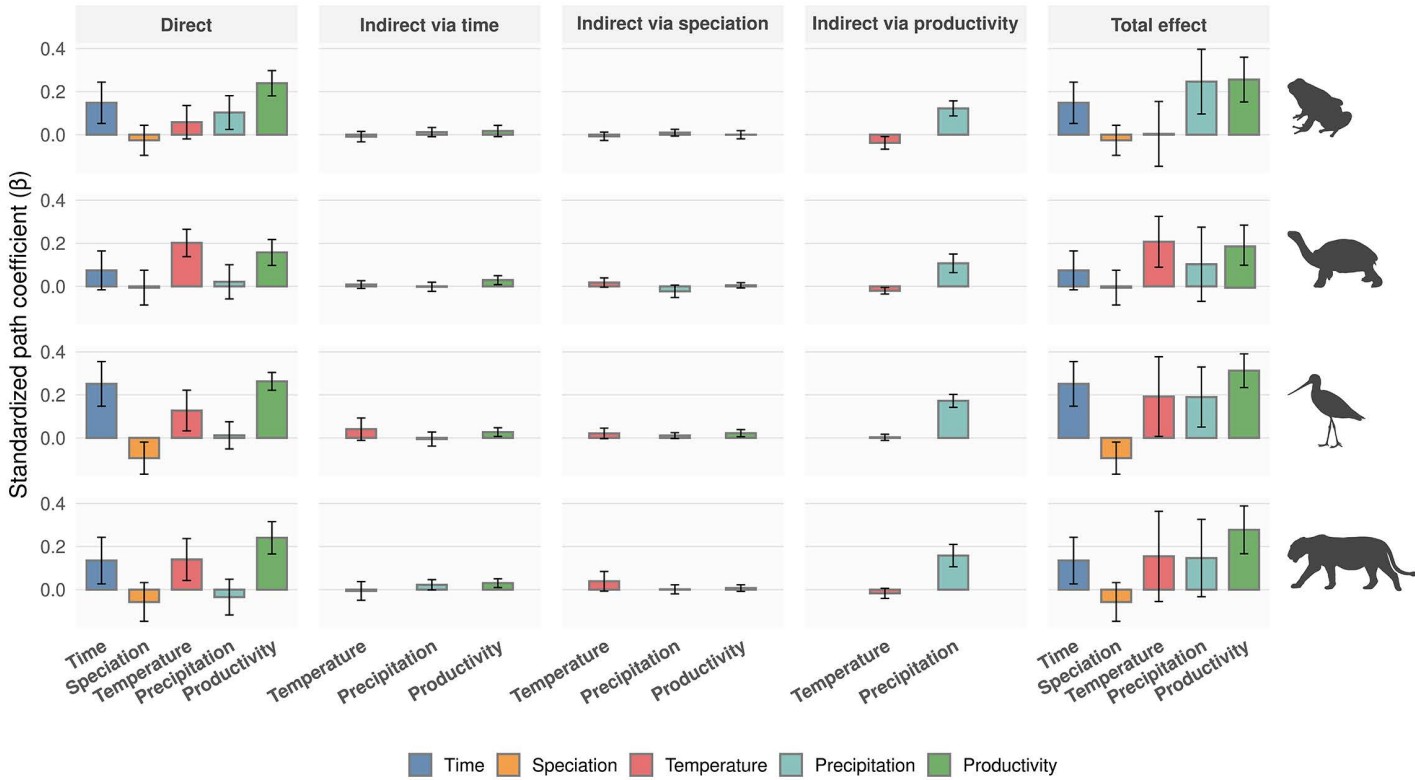

**Fig 4. Direct, indirect, and total effects of evolutionary time, speciation rates, and environmental factors on species richness across tetrapod clades.** Bars represent mean standardized path coefficients (β) estimated across clades for amphibians (34 clades), reptiles (34 clades), birds (31 clades), and mammals (30 clades). Error bars indicate 95% confidence intervals across clades. The five predictors are evolutionary time (assemblage age), speciation rate (DR estimates), temperature, precipitation, and net primary productivity (NPP). Direct effects represent the standardized path coefficients linking each predictor to species richness. Indirect effects are grouped into three classes: environmental effects mediated via evolutionary time, environmental effects mediated via speciation rates, and climatic effects mediated via productivity. Total effects represent the sum of direct and all indirect pathways linking each predictor to species richness. Colours correspond to predictor variables included in the structural equation models. Variation in effect sizes among individual clades is shown in O–R in S1 Appendix Figs. Silhouette images are reproduced under their original licenses and were obtained from PhyloPic (https://www.phylopic.org). Mammal silhouette (*Panthera leo*) by Gabriela Palomo-Munoz (CC BY 4.0); reptile silhouette (*Megalochelys atlas*) by Roberto Díaz Sibaja (CC BY 4.0); bird silhouette (*Limosa fedoa*) and amphibian silhouette (*Phyllobates terribilis*) by Andy Wilson (CC0 1.0 Universal, public domain). The data and R scripts used to generate this figure are publicly available in the Zenodo repository (DOI: https://doi.org/10.5281/zenodo.14008084).

### Direct and indirect effects on species richness

Productivity emerged as the strongest and most consistent driver of species richness, exerting positive direct effects in all tetrapod classes (Figs 4; O in S1 Appendix). Evolutionary time showed moderate direct effects overall, being strongest in birds, intermediate in amphibians and mammals, and weak in reptiles (Figs 4; O in S1 Appendix). Speciation rates showed weak or negligible effects overall, with the most pronounced negative estimates occurring in birds (Figs 4; O in S1 Appendix). Climatic variables displayed more heterogeneous direct effects. Temperature showed generally positive, moderate effects across clades, with comparatively stronger effects in reptiles than in other tetrapod groups (Figs 4; O in S1 Appendix). Precipitation had overall weak direct effects, although these were relatively stronger in amphibians (Figs 4; O in S1 Appendix). Together, these results indicate variation among tetrapod classes in the strength and direction of climatic effects on species richness.

Indirect effects of climate on species richness were dominated by pathways mediated through productivity (Figs 4; P in S1 Appendix). Specifically, precipitation influenced species richness primarily indirectly via productivity across clades, whereas temperature had consistently weak indirect effects through productivity (Figs 4; P in S1 Appendix). Indirect pathways linking environmental variables (productivity, temperature, and precipitation) to species richness through evolutionary time or speciation rates were generally weak and inconsistent across clades (Figs 4; Q–R in S1 Appendix). Collectively, these results indicate that the principal indirect route connecting climate to species richness operates through productivity, while alternative indirect mechanisms involving evolutionary time or diversification contribute little to large-scale richness patterns.

We also tested separate SEMs representing each of the four richness-generating scenarios (Fig 2) and evaluated them via model-selection (Figs 5; S–V in S1 Appendix). Comparisons using both AIC and BIC consistently favored equilibrium-based dynamics across tetrapods (Figs 5; S–V in S1 Appendix), with the *Stable persistence* scenario supported in ~71% of clades (92 of 129), whereas *Historical legacy* received more limited support (≈11%), and diversification-driven scenarios (*Cradles of diversity* and *Speciation balance*) were rarely supported (≈9% each). Together, these results indicate that spatial variation in species richness across tetrapod clades is primarily determined by contemporary productivity rather than by differences in diversification rates or evolutionary time. Overall, the SEMs explained a substantial fraction of richness variation (mean clade-level $R^2 = 36\%$–46% across classes; Fig W in S1 Appendix), and model fit was generally acceptable across clades (Figs X–AA in S1 Appendix).

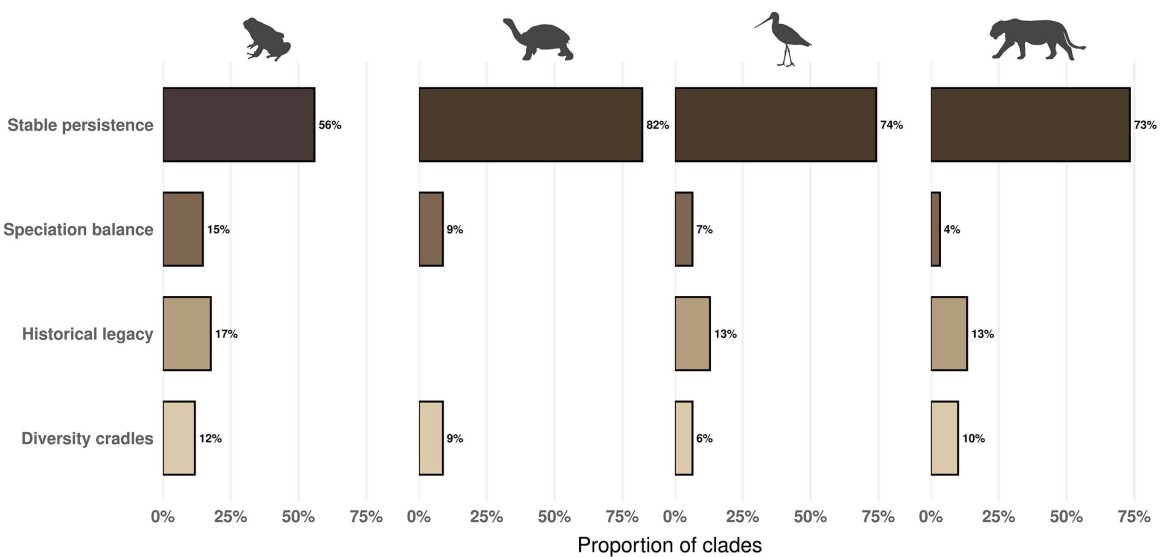

**Fig 5. Support for alternative richness-generating scenarios across tetrapod clades.** Bars show the proportion of clades within each tetrapod class (amphibians, reptiles, birds, and mammals) for which each richness-generating scenario received the strongest support based on Akaike Information Criterion (AIC). Each clade was fitted independently with four alternative SEMs corresponding to the *Stable persistence*, *Speciation balance*, *Historical legacy*, and *Cradles of diversity* scenarios (Fig 2), and support was assigned to the scenario with the lowest AIC. Across all tetrapod classes, equilibrium-based dynamics, particularly the *Stable persistence* scenario, were most frequently supported, whereas nonequilibrium scenarios received comparatively less support. Results based on Bayesian Information Criterion (BIC) were identical and are presented in the S–V in S1 Appendix Figs. Silhouette images are reproduced under their original licenses and were obtained from PhyloPic (https://www.phylopic.org). Mammal silhouette (*Panthera leo*) by Gabriela Palomo-Munoz (CC BY 4.0); reptile silhouette (*Megalochelys atlas*) by Roberto Díaz Sibaja (CC BY 4.0); bird silhouette (*Limosa fedoa*) and amphibian silhouette (*Phyllobates terribilis*) by Andy Wilson (CC0 1.0 Universal, public domain). The data and R scripts used to generate this figure are publicly available in the Zenodo repository (DOI: https://doi.org/10.5281/zenodo.14008084).

## Influence of clade traits on richness-generating processes

We evaluated whether the direct effects of environmental variables, speciation rates, and evolutionary time on species richness varied systematically with clade thermal physiology, age, diversity (species richness), and geographic extent (number of grid cells occupied). Clade age was negatively correlated only with the strength of evolutionary time effects (Figs 6; AB in S1 Appendix), indicating that younger clades show stronger influence of evolutionary time on species richness. Clade thermal physiology did not significantly affect variation in any direct effect (all $p > 0.05$), suggesting that ectothermic and endothermic clades are influenced by evolutionary time, speciation, and environmental factors in broadly similar ways across tetrapods (Fig AC in S1 Appendix). Similarly, clade diversity and geographic extent were not significantly associated with the strength of any richness-driver effects (all $p > 0.05$; Figs AD–AE in S1 Appendix), indicating that neither the total number of species in a clade nor the spatial breadth of its distribution systematically alters the relative importance of evolutionary time, speciation, or environment in influencing species richness.

## Robustness analyses

We evaluated the robustness of our results to alternative methodological choices, data sources, and potential sources of bias. First, varying the number of Laplacian spectral modalities (40, 50, and 60) or excluding paraphyletic clades, which represented 11% of all modalities, produced effect sizes for richness predictors that were qualitatively similar to the main analyses (Figs AF–AH in S1 Appendix). Similarly, using molecular-only phylogenies, which excluded species with taxonomically imputed positions, produced estimates of evolutionary time, speciation rates, and environmental effects consistent with those from the full phylogenies (Figs AI–AL in S1 Appendix). These results indicate that our conclusions are robust to variation in clade delineation and phylogenetic data sources.

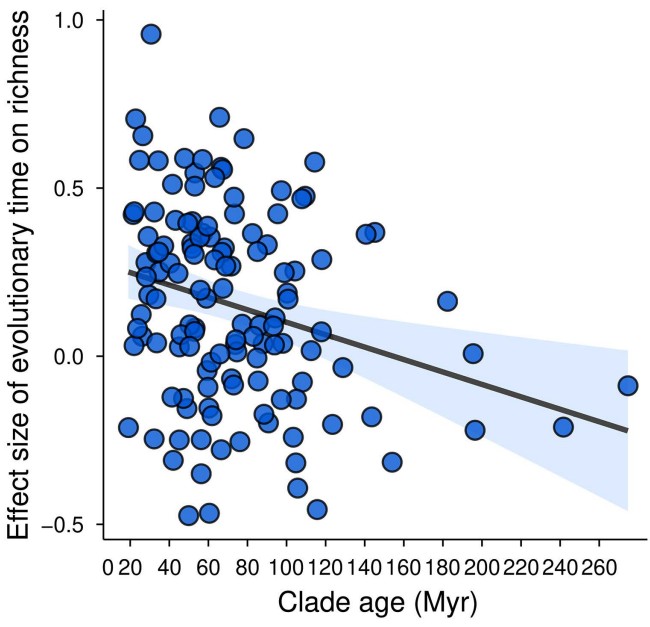

**Fig 6. Effect of clade crown age on the strength of the relationship between species richness and evolutionary time (assemblage age) across tetrapod clades.** The solid line shows the fitted linear regression, with the shaded band indicating the 95% confidence interval. Only this relationship involving clade age was statistically supported. Other evaluated clade traits, including thermal physiology, clade diversity, and geographic extent, showed no significant effects and are reported in AB–AE in S1 Appendix Figs. The data and R scripts used to generate this figure are publicly available in the Zenodo repository (DOI: https://doi.org/10.5281/zenodo.14008084).

Second, we evaluated whether our conclusions depended on the choice of evolutionary time metric. Assemblage age showed weak or no correlation with MPD and MBL across tetrapod clades (Figs AM–AP in S1 Appendix), indicating that these metrics capture distinct aspects of evolutionary history. When MPD or MBL were used as predictors in the SEMs, the estimated effects of evolutionary time on species richness remained qualitatively similar in direction to those obtained with assemblage age (Figs AQ and AR in S1 Appendix). These analyses are treated strictly as robustness checks, with all main inferences based on the biogeography-derived assemblage age metric.

Third, we evaluated whether our conclusions regarding speciation contributions to species richness were sensitive to the choice of tip-rate metric. Speciation rates derived from DR, BAMM, and ClaDS were strongly positively correlated and produced consistent spatial patterns across clades (Figs AS–AV in S1 Appendix). Consequently, all three approaches led to the same qualitative inferences regarding the contribution of speciation to species richness, confirming the robustness of our results to the method of speciation rate estimation. Finally, although residuals of the clade-specific SEMs exhibited strong spatial autocorrelation (Fig H in S1 Appendix), accounting for spatial structure did not alter inferred richness drivers. Re-fitting all models with latitude and longitude included as predictors resulted in only minor quantitative changes in coefficients (Fig AW in S1 Appendix).

## Discussion

By applying a conceptual framework that distinguishes four richness-generating scenarios (Figs 1 and 2), our study provides an integrated assessment of how evolutionary time, diversification dynamics, climate, and environmental energy jointly influence global patterns of species richness across tetrapod clades. We found that productivity showed strong, consistent direct effects on species richness across clades, largely independent of clade traits such as age, physiology, diversity, or geographic extent (Figs 4; O in S1 Appendix). Precipitation contributed indirectly to richness primarily by influencing productivity, further highlighting the central role of environmental energy in governing richness patterns. Other factors played more limited or context-dependent roles: evolutionary time and temperature influenced richness in some clades but with weaker and less consistent effects, whereas speciation rates contributed little to explaining richness patterns (Figs 4; O in S1 Appendix). These results were robust across a wide range of sensitivity analyses, including alternative Laplacian clade delineations, exclusion of taxonomically imputed species, and the use of different diversification metrics (Figs AF–AV in S1 Appendix). Taken together, these results provide strong support for equilibrium-based explanations, in which environmental factors, particularly energy availability, regulate species richness. Nevertheless, deviations from these patterns in younger clades highlight the influence of historical context on species richness.

### Equilibrium dynamics as a primary explanation of species richness patterns

Our results are most consistent with the *Stable persistence* scenario, which received substantially greater support than alternative richness-generating scenarios across tetrapod clades (Figs 5; S–V in S1 Appendix). This indicates that species richness reflects a dynamic equilibrium between speciation and extinction, primarily regulated by productivity, in accord with the Equilibrium Theory of Biodiversity Dynamics (ETBD) [35,36,90]. Under this perspective, productivity does not act as a hard ceiling on richness. Instead, by modulating demographic stability and population sizes, it likely affects persistence times and extinction rates [82], thereby influencing the long-term balance between speciation and extinction rates and resulting turnover rates (Fig 7). A key implication of this framework is that speciation rates need not be positively associated with standing species richness. Consistent with this, we find that speciation rates across tetrapod clades are weakly, and often negatively, related to richness (Figs 4; O in S1 Appendix), echoing a well-documented macroevolutionary paradox in which regions with high richness typically reveal lower speciation rates [15,25,27,112,113].

This apparent decoupling of speciation and richness can be reconciled by considering the joint roles of extinction and ecological opportunity. Ecological opportunity, arising from low competition, unoccupied niche space, or recent

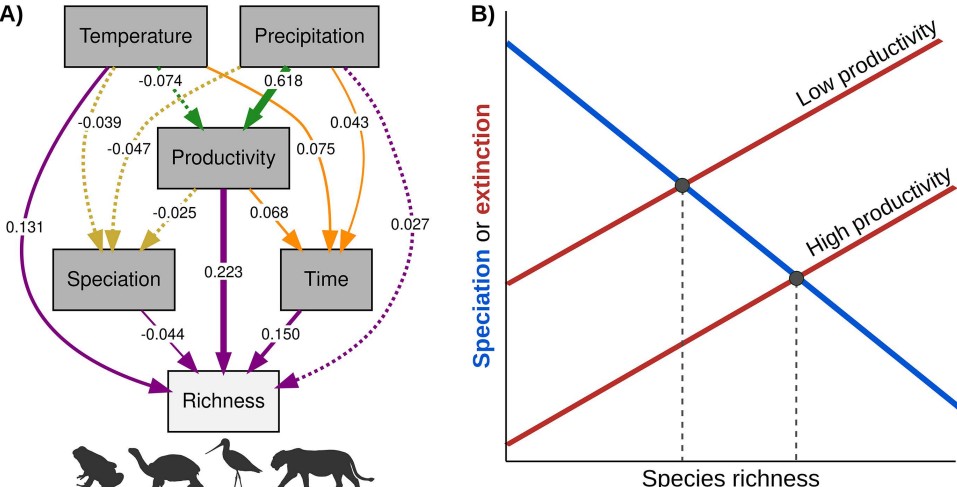

**Fig 7. Synthesis of structural equation model results across tetrapod clades and their interpretation within an equilibrium species richness framework. A)** Path diagram showing mean standardized path coefficients (β) for direct and indirect effects on species richness. Arrow colors denote different pathways: direct effects on richness (purple), indirect effects mediated by evolutionary time (orange), speciation rate (golden yellow), and productivity (green). Arrow width is proportional to effect strength. Dashed arrows indicate nonsignificant effects, defined as 95% confidence intervals overlapping zero. **B)** Conceptual scenario consistent with our findings. Regional differences in productivity generate variation in extinction levels, represented by distinct extinction curves (red lines). Speciation is assumed to be diversity-dependent with a consistent dependence across regions (blue line), such that per-species speciation rates decline with increasing species richness (e.g., higher species richness leads to lower population sizes, decreasing speciation probability) [35]. Equilibrium diversity levels emerge where speciation and extinction balance, producing a negative association between species richness and speciation rate [90]. The specific shape or slope of the extinction curves does not matter here (extinction may even be diversity-independent), as long as extinction propensity differ between regions. Silhouette images are reproduced under their original licenses and were obtained from PhyloPic (https://www.phylopic.org). Mammal silhouette (*Panthera leo*) by Gabriela Palomo-Munoz (CC BY 4.0); reptile silhouette (*Megalochelys atlas*) by Roberto Díaz Sibaja (CC BY 4.0); bird silhouette (*Limosa fedoa*) and amphibian silhouette (*Phyllobates terribilis*) by Andy Wilson (CC0 1.0 Universal, public domain). The data used to generate the panel A are publicly available in the Zenodo repository (DOI: https://doi.org/10.5281/zenodo.14008084).

environmental change, can promote rapid speciation, particularly in species-poor or climatically dynamic regions [114,115]. Such conditions are often characteristic of temperate or recently disturbed regions, where newly available niches may facilitate diversification even when long-term persistence is limited [115,116]. However, ecological opportunity is inherently transient. In regions characterized by high climatic variability or low-energy availability, elevated extinction risk can counterbalance gains from speciation, preventing the accumulation of high standing richness despite ongoing diversification [26,90,117–119].

Under this interpretation, high speciation rates in temperate regions reflect rapid lineage turnover rather than sustained species accumulation (Fig 7). Although extinction rates are difficult to estimate reliably from molecular phylogenies alone [94,99,103], multiple lines of evidence indicate that extinction probabilities are elevated in climatically unstable and low-energy environments [8,120–122]. Consequently, these regions may experience bursts of diversification without corresponding increases in standing richness, despite high speciation rates [115,116]. Taken together, this equilibrium perspective reconciles two central patterns observed across tetrapods: the lack of a positive relationship between recent speciation rates and species richness, and the consistently strong positive relationship between productivity and richness patterns. Ecological opportunity may promote speciation, but long-term richness is ultimately governed by energy-dependent constraints on persistence and extinction.

## The context-dependent roles of evolutionary time and climate

The influence of evolutionary time on species richness varied markedly among tetrapod clades (Figs 4; O in S1 Appendix), with the strength and direction of this effect depending on clade age (Fig 6). Specifically, the relationship between clade

age and the strength of evolutionary time effects on species richness followed a triangular pattern: older clades consistently show small effect sizes, whereas younger clades show a wide range of effect sizes (Fig 6). This empirical pattern mirrors predictions from eco-evolutionary simulation models [16], which suggest that time-for-speciation effects dominate richness patterns early in clade history. Consistent with this expectation, younger clades differ in how far they have progressed toward equilibrium, resulting in substantial heterogeneity in the importance of evolutionary time [19,38,49,50]. By contrast, in older clades ecological limits (i.e., diversity equilibria) are likely to have been approached or reached, reducing the extent to which additional time translates into further species accumulation [4,5,42,123]. Importantly, other clade-level attributes, including physiology, total diversity, and geographic extent, showed no systematic influence on the magnitude or direction of richness–driver relationships (Figs AC–AE in S1 Appendix).

Climatic variables also showed context-dependent effects, but through distinct pathways. Precipitation affected species richness mainly through indirect pathways mediated by productivity (Fig 4). This pattern is consistent with the idea that water availability regulates richness mainly by enhancing primary production, rather than acting as a direct constraint on most tetrapod clades [58,59,124]. An important exception arises in amphibians, where precipitation exerts a notably stronger total effect on richness than in other tetrapod clades (Fig 4). This occurs because precipitation influences amphibians both indirectly via productivity and directly by constraining water availability, which is critical for their physiology and reproduction [125,126]. The lack of comparable direct precipitation effects in other tetrapods indicates that water availability constrains species richness directly only in lineages whose life histories and reproductive success are tightly coupled to environmental moisture, whereas in most tetrapods precipitation influences richness primarily through its effects on ecosystem productivity.

Temperature, on the other hand, presented a more puzzling pattern. Although it exerted consistent direct effects on species richness, particularly in reptiles, these effects were not mediated by productivity, evolutionary time, or speciation rates (Fig 4), which are the pathways most commonly invoked to explain temperature–diversity relationships [28,37,45,47,56]. In reptiles, however, this effect is readily interpretable in light of the central role of temperature in ectothermic physiology. Temperature directly constrains metabolic performance, activity time, growth, and reproductive output [127,128], which has long been invoked to explain broad-scale gradients in reptile species richness [46,57,129]. However, the observed direct effects of temperature, which comprises not only reptiles, suggest a broader interpretation. Temperature may act as a surrogate for aspects of resource availability that are not fully captured by NPP, particularly for higher-trophic-level consumers [31,130]. Most tetrapods are not primary consumers: amphibians lack herbivorous species entirely [125], reptiles include relatively few herbivores [131], and herbivory in birds is largely seed-based rather than foliar [132]. Consequently, the resources limiting tetrapod diversity may be more closely tied to the biomass, abundance, and productivity of animal prey than to plant production per se. Because energy transfer across trophic levels is strongly nonlinear [133], and MODIS-based NPP may not perfectly reflect real resource abundance [134], variation in measured NPP can translate only imperfectly into the energy available to these consumers. Under this interpretation, the direct effects of temperature do not contradict the dominant role of productivity. They rather complement it by reinforcing an equilibrium perspective in which species richness is regulated by resource availability, rather than through diversification or time-dependent processes.

## Methodological considerations, limitations, and robustness

Our conclusions rest on a set of explicit methodological choices that warrant careful justification. First, inference on diversification dynamics is inherently constrained by the properties of available metrics. Tip-based speciation rate estimators reliably capture recent diversification processes, but they provide limited information about deep-time dynamics [99–101]. Estimating extinction rates from timetrees remains particularly problematic, as extinction leaves weak and often unidentifiable signatures in molecular phylogenies [94,103]. To account for these limitations, we relied on multiple complementary speciation metrics, which consistently yielded congruent patterns across clades (Figs AS–AV in S1 Appendix). This

agreement increases confidence that our main conclusions regarding the limited role of speciation on species richness are robust to the specific choice of metric and reflect genuine macroecological patterns rather than methodological artifacts. Although extinction was not estimated directly, our interpretations were informed by well-established theoretical expectations under equilibrium dynamics [35,90]. In particular, extinction is thought to be elevated in low-energy or climatically unstable regions [1,90,115,117], allowing us to interpret broad geographic patterns consistent with elevated turnover (e.g., in temperate regions; Fig 7) even in the absence of explicit extinction estimates.

Second, quantifying evolutionary time poses long-standing challenges in macroecological analyses. Widely used phylometric proxies such as MPD and MBL conflate lineage age with diversification dynamics and do not explicitly account for dispersal history [90], limiting their ability to represent the time available for regional species accumulation. To address these limitations, we employed an assemblage age metric derived from ancestral range reconstructions, which estimates the timing of lineage arrival into regional assemblages [86–88]. This measure provides an approximation of the evolutionary window during which lineages have been present in a region and may have contributed to local richness. While this approach does not eliminate uncertainty associated with historical biogeographic inference, it allows dispersal history to be considered explicitly and offers a complementary perspective to traditional phylometric measures when evaluating the role of evolutionary time on species richness.

Third, a key methodological feature of our study is the use of Laplacian spectral partitioning to delineate clades [55]. Unlike approaches that rely on taxonomic ranks or arbitrary node sampling [77], this data-driven method partitions the phylogeny based on the spectral decomposition of its Laplacian matrix, grouping species according to shared branching structure. This procedure yields nonoverlapping clades defined by an explicit optimization criterion applied uniformly across the tree, ensuring objective and reproducible clade delineation. As a result, it represents an important advance over previous macroecological studies, because it enables true replication of richness–driver analyses across many independent evolutionary lineages that differ in their diversification dynamics. At the same time, because Laplacian partitioning is derived from phylogenetic topology and branch lengths, it could in principle be influenced by phylogenetic uncertainty, by the inclusion of paraphyletic clades, or by the choice of spectral resolution (i.e., the number of modalities). To explicitly evaluate these possibilities, we conducted targeted sensitivity analyses. Specifically, we repeated all analyses after excluding paraphyletic clades (~11% of all clades), varied the number of Laplacian modalities over a broad range (40–60), and restricted the analyses to clades derived from molecular-only phylogenies. Across all these tests, effect sizes, scenario support, and qualitative conclusions remained highly consistent (Figs AF–AL in S1 Appendix), indicating that our results are robust to potential biases associated with clade delineation and phylogenetic structure.

Finally, we acknowledge potential sources of circularity arising from shared data inputs. Two possibilities merit attention. First, clade delineation via the Laplacian spectrum groups species based on shared branching structure, which could in principle result in clades with more similar tip-based speciation rates than expected by chance. However, if this were driving our results, altering the spectral resolution or clade composition would be expected to change effect sizes or scenario support. This was not the case, whereby varying the number of Laplacian modalities (40–60), excluding paraphyletic clades, and restricting analyses to molecular-only phylogenies all yielded consistent results (Figs AF–AL in S1 Appendix). Second, species range maps underpin both ancestral range reconstructions and species richness estimates, and in some cases have been interpolated using climatic variables such as temperature or precipitation. This raises the possibility that assemblage age or climate–richness relationships could be partially influenced by shared spatial information. However, several considerations argue against this. Species richness is calculated independently as a count of species per grid cell, net primary productivity (the strongest predictor in our models) was not used in range maps interpolation, and climatic variables did not emerge as dominant drivers of richness. If any circularity were present, it would be expected to inflate climate effects rather than productivity effects, making our conclusions regarding the primacy of productivity conservative.

Across all potential sources of uncertainty, such as diversification metrics, representations of evolutionary time, clade delineation, and data circularity, our conclusions remained consistent. Taken together, these findings indicate that support

for productivity-driven, equilibrium-based explanations of species richness across tetrapod clades reflects a robust and general macroecological signal, rather than sensitivity to particular assumptions or methodological choices. By combining replicated clade-level analyses with models that explicitly incorporate historical, diversification, and environmental factors, our framework allows disentangling the relative contributions of these processes to large-scale diversity patterns.

## Conclusions

The drivers of species richness patterns have been controversial and notoriously hard to tease apart, with decades of research yielding seemingly contradictory results. Much of this inconsistency stems from conceptual ambiguity in how the three classic explanations (evolutionary time, diversification rates, and ecological limits) are defined and evaluated, as they operate at different levels and are not mutually exclusive. To address this problem, we introduced a unifying conceptual framework that distinguishes richness-generating processes based on two orthogonal dimensions: the role of evolutionary time and speciation rates. This formulation subsumes traditional hypotheses, accommodates both equilibrium and nonequilibrium dynamics, and yields four baseline scenarios with explicit, testable predictions.

Applying this framework across tetrapod clades, we find strong and consistent support for an equilibrium-based explanation of species richness regulated primarily by productivity. Net primary productivity emerged as the dominant predictor of richness across clades, whereas recent speciation rates contributed little to species richness, reflecting lineage turnover rather than long-term accumulation of lineages. The influence of evolutionary time and temperature was more contingent. Evolutionary time had a stronger effect on species richness in younger clades, consistent with these clades not yet having reached equilibrium, whereas its influence was weak in older clades. Temperature, in contrast, exerted predominantly direct effects on richness that were not mediated by time, diversification, or productivity. These results may suggest that temperature functions as a more effective proxy for resource availability across trophic levels than productivity in many tetrapod clades, particularly those relying on animal prey rather than primary plant biomass.

By distinguishing alternative richness-generating scenarios and testing their predictions across many independently defined clades, our study helps reconcile long-standing debates over the relative roles of evolutionary time, diversification, and ecological limits in governing species richness patterns. More broadly, our results indicate that diversity gradients across tetrapods are best understood as the outcome of equilibrium dynamics regulated by resource availability, with evolutionary time and climate shaping richness in context-dependent and lineage-specific ways. This framework provides a coherent foundation for future comparative studies seeking to disentangle the processes underlying biodiversity patterns across the Tree of Life.

## Supporting information

**S1 Appendix. Supplementary figures and tables. A Fig.** Species richness patterns for (a) amphibians ($n = 5{,}235$ species), (b) reptiles ($n = 9{,}129$ species), (c) birds ($n = 9{,}324$ species), and (d) mammals ($n = 5{,}145$ species). Species richness was estimated by aggregating species distribution range maps into equal-area grid cells at a spatial resolution of ~1° × 1°, and summing the number of species whose geographic ranges overlapped each grid cell. The maps shown here illustrate overall richness patterns for each tetrapod class, however, all subsequent analyses were conducted on species richness patterns calculated separately for each focal clade. Continental boundaries were obtained from Natural Earth Admin 0 Country Boundaries (https://www.naturalearthdata.com), which is in the public domain (CC0) and compatible with the CC BY 4.0 license. The map was generated in R using this base layer. The data underlying this figure can be found in https://doi.org/10.5281/zenodo.14008084. **B Fig.** Global spatial distribution of environmental predictors used in the analyses: (a) mean annual temperature, (b) mean annual precipitation, and (c) net primary productivity (NPP). Continental boundaries were obtained from Natural Earth Admin 0 Country Boundaries (https://www.naturalearthdata.com), which is in the public domain (CC0) and compatible with the CC BY 4.0 license. The map was generated in R using this base layer. The data underlying this figure can be found in https://doi.org/10.5281/zenodo.14008084. **C Fig.** Conceptual illustration of

the workflow used to estimate evolutionary time (assemblage age) across tetrapod clades. The example shows a hypothetical clade and summarizes the main analytical steps. (a) Spatial variation in species richness across grid cells within the clade. (b) Phylogenetic turnover among assemblages is quantified using species distributions and phylogenetic relationships to delineate evolutionary regions defined by shared evolutionary history. (c) Ancestral geographic ranges are reconstructed along the phylogeny using the DEC model implemented in BioGeoBEARS, yielding probabilistic estimates of ancestral area for each node. (d) For each species present in a given grid cell, the phylogeny is traced back to the deepest ancestral node whose reconstructed range includes the focal evolutionary region. The age of this node is interpreted as the species' arrival time into the assemblage (grid cell). Assemblage age is then calculated as the mean arrival time across all species occurring within a grid cell. Continental boundaries were obtained from Natural Earth Admin 0 Country Boundaries (https://www.naturalearthdata.com), which is in the public domain (CC0) and compatible with the CC BY 4.0 license. The map was generated in R using this base layer. The data underlying this figure can be found in https://doi.org/10.5281/zenodo.14008084. **D Fig.** Variance inflation factors (VIF) for predictors of species richness across amphibian clades. Violins show the distribution of VIF values across clades for each predictor, with embedded boxplots indicating medians and interquartile ranges. Points represent individual clades. Predictors include evolutionary time (assemblage age), speciation rate (DR), temperature, precipitation, and productivity (NPP). Dashed horizontal lines indicate VIF thresholds of 5 (moderate multicollinearity) and 10 (high multicollinearity). The data underlying this figure can be found in https://doi.org/10.5281/zenodo.14008084. **E Fig.** Variance inflation factors (VIF) for predictors of species richness across reptile clades. Violins show the distribution of VIF values across clades for each predictor, with embedded boxplots indicating medians and interquartile ranges. Points represent individual clades. Predictors include evolutionary time (assemblage age), speciation rate (DR), temperature, precipitation, and productivity (NPP). Dashed horizontal lines indicate VIF thresholds of 5 (moderate multicollinearity) and 10 (high multicollinearity). The data underlying this figure can be found in https://doi.org/10.5281/zenodo.14008084. **F Fig.** Variance inflation factors (VIF) for predictors of species richness across bird clades. Violins show the distribution of VIF values across clades for each predictor, with embedded boxplots indicating medians and interquartile ranges. Points represent individual clades. Predictors include evolutionary time (assemblage age), speciation rate (DR), temperature, precipitation, and productivity (NPP). Dashed horizontal lines indicate VIF thresholds of 5 (moderate multicollinearity) and 10 (high multicollinearity). The data underlying this figure can be found in https://doi.org/10.5281/zenodo.14008084. **G Fig.** Variance inflation factors (VIF) for predictors of species richness across mammal clades. Violins show the distribution of VIF values across clades for each predictor, with embedded boxplots indicating medians and interquartile ranges. Points represent individual clades. Predictors include evolutionary time (assemblage age), speciation rate (DR), temperature, precipitation, and productivity (NPP). Dashed horizontal lines indicate VIF thresholds of 5 (moderate multicollinearity) and 10 (high multicollinearity). The data underlying this figure can be found in https://doi.org/10.5281/zenodo.14008084. **H Fig.** Spatial autocorrelation of residuals from clade-specific path-models across tetrapod classes. Bars show Moran's I values calculated for model residuals within each clade for amphibians, reptiles, birds, and mammals. The data underlying this figure can be found in https://doi.org/10.5281/zenodo.14008084. **I Fig.** Amphibian phylogeny showing clades delineated using the Laplacian spectrum approach. This procedure identified 34 amphibian clades with more than 50 species each, totaling 6,361 species. Clades with fewer than 50 species were excluded from downstream analyses. Colours and associated numbers denote distinct clades. Numbers highlighted in color (e.g., clade 31) indicate paraphyletic clades. Paraphyletic clades represent a minority of cases, and their exclusion did not alter the main results. The data underlying this figure can be found in https://doi.org/10.5281/zenodo.14008084. **J Fig.** Reptile phylogeny showing clades delineated using the Laplacian spectrum approach. The analysis identified 34 reptile clades with more than 50 species each, comprising 32 squamate clades (left panel) and 2 turtle–crocodilian clades (right panel), totaling 9,081 species. Clades with fewer than 50 species were excluded from downstream analyses. Colours and associated numbers denote distinct clades. Numbers highlighted in color (e.g., clade 25) indicate paraphyletic clades. Paraphyletic clades were rare, and their exclusion did not affect the main conclusions.

The data underlying this figure can be found in https://doi.org/10.5281/zenodo.14008084. **K Fig.** Bird phylogeny showing clades delineated using the Laplacian spectrum approach. This method identified 31 bird clades with more than 50 species each, totaling 9,232 species. Clades with fewer than 50 species were excluded from downstream analyses. Colours and associated numbers denote distinct clades. Numbers highlighted in color (e.g., clade 3) indicate paraphyletic clades. These clades represent a small fraction of the dataset and do not influence the overall results. The data underlying this figure can be found in https://doi.org/10.5281/zenodo.14008084. **L Fig.** Mammal phylogeny showing clades delineated using the Laplacian spectrum approach. The analysis identified 30 mammal clades with more than 50 species each, totaling 5,032 species. Clades with fewer than 50 species were excluded from downstream analyses. Colours and associated numbers denote distinct clades. Numbers highlighted in color (e.g., clade 7) indicate paraphyletic clades. The exclusion of these clades did not change the main findings. The data underlying this figure can be found in https://doi.org/10.5281/zenodo.14008084. **M Fig.** Variation in species richness and crown age across tetrapod clades. Each circle represents an individual clade, with colors indicating tetrapod class (34 amphibian clades, 34 reptile clades, 31 bird clades, and 30 mammal clades). Marginal density plots show the distributions of species richness and clade age within each tetrapod class. The data underlying this figure can be found in https://doi.org/10.5281/zenodo.14008084. **N Fig.** Geographic extent of terrestrial tetrapod clades included in the analyses. (a) Density distributions of clade geographic extent (range size) for amphibians, reptiles, birds, and mammals. (b) Spatial distribution of the smallest clade geographic extent observed, corresponding to an amphibian clade. (c) Spatial distribution of the largest clade geographic extent observed, corresponding to a bird clade. Continental boundaries were obtained from Natural Earth Admin 0 Country Boundaries (https://www.naturalearthdata.com), which is in the public domain (CC0) and compatible with the CC BY 4.0 license. The map was generated in R using this base layer. The data underlying this figure can be found in https://doi.org/10.5281/zenodo.14008084. **O Fig.** Direct effects of environmental factors (temperature, precipitation, and productivity), evolutionary time (assemblage age), and speciation rate on species richness across tetrapod clades. Points represent mean standardized path coefficients (β) across clades, with error bars indicating 95% confidence intervals, shown separately for amphibians, reptiles, birds, and mammals. Black diamonds indicate the mean effect size of each predictor across clades within each tetrapod class. Colours denote effect direction and statistical significance: blue, positive and significant; red, negative and significant; green, nonsignificant (confidence intervals overlapping zero). The data underlying this figure can be found in https://doi.org/10.5281/zenodo.14008084. **P Fig.** Indirect effects of climate mediated through productivity on species richness across tetrapod clades. Points represent mean standardized path coefficients (β) across clades, with error bars indicating 95% confidence intervals, shown separately for amphibians, reptiles, birds, and mammals. Black diamonds indicate the mean effect size of each predictor across clades within each tetrapod class. Colours denote effect direction and statistical significance: blue, positive and significant; red, negative and significant; green, nonsignificant (confidence intervals overlapping zero). The data underlying this figure can be found in https://doi.org/10.5281/zenodo.14008084. **Q Fig.** Indirect effects of environmental factors mediated through evolutionary time (assemblage age) on species richness across tetrapod clades. Points represent mean standardized path coefficients (β) across clades, with error bars indicating 95% confidence intervals, shown separately for amphibians, reptiles, birds, and mammals. Black diamonds indicate the mean effect size of each predictor across clades within each tetrapod class. Colours denote effect direction and statistical significance: blue, positive and significant; red, negative and significant; green, nonsignificant (confidence intervals overlapping zero). The data underlying this figure can be found in https://doi.org/10.5281/zenodo.14008084. **R Fig.** Indirect effects of environmental factors mediated through speciation rate on species richness across tetrapod clades. Points represent mean standardized path coefficients (β) across clades, with error bars indicating 95% confidence intervals, shown separately for amphibians, reptiles, birds, and mammals. Black diamonds indicate the mean effect size of each predictor across clades within each tetrapod class. Colours denote effect direction and statistical significance: blue, positive and significant; red, negative and significant; green, nonsignificant (confidence intervals overlapping zero). The data underlying this figure can be found in https://doi.org/10.5281/

zenodo.14008084. **S Fig.** Model-selection support for alternative richness-generating scenarios across amphibian clades based on information-theoretic criteria. Heatmaps show (a) ΔBIC and (b) ΔAIC values for four scenario-specific path-models (stable persistence, historical legacy, diversity cradles, and speciation balance) evaluated separately for each clade (rows). Lower Δ values indicate stronger relative model support within a clade. For each clade, the scenario with the lowest information-criterion value is outlined with a darker border, indicating the most strongly supported model. Colors represent relative ΔBIC or ΔAIC values within clades. The data underlying this figure can be found in https://doi. org/10.5281/zenodo.14008084. **T Fig.** Model-selection support for alternative richness-generating scenarios across reptile clades based on information-theoretic criteria. Heatmaps show (a) ΔBIC and (b) ΔAIC values for four scenario-specific path-models (stable persistence, historical legacy, diversity cradles, and speciation balance) evaluated separately for each clade (rows). Lower Δ values indicate stronger relative model support within a clade. For each clade, the scenario with the lowest information-criterion value is outlined with a darker border, indicating the most strongly supported model. Colors represent relative ΔBIC or ΔAIC values within clades. The data underlying this figure can be found in https://doi. org/10.5281/zenodo.14008084. **U Fig.** Model-selection support for alternative richness-generating scenarios across bird clades based on information-theoretic criteria. Heatmaps show (a) ΔBIC and (b) ΔAIC values for four scenario-specific path models (stable persistence, historical legacy, diversity cradles, and speciation balance) evaluated separately for each clade (rows). Lower Δ values indicate stronger relative model support within a clade. For each clade, the scenario with the lowest information-criterion value is outlined with a darker border, indicating the most strongly supported model. Colors represent relative ΔBIC or ΔAIC values within clades. The data underlying this figure can be found in https://doi. org/10.5281/zenodo.14008084. **V Fig.** Model-selection support for alternative richness-generating scenarios across mammal clades based on information-theoretic criteria. Heatmaps show (a) ΔBIC and (b) ΔAIC values for four scenario-specific path models (stable persistence, historical legacy, diversity cradles, and speciation balance) evaluated separately for each clade (rows). Lower Δ values indicate stronger relative model support within a clade. For each clade, the scenario with the lowest information-criterion value is outlined with a darker border, indicating the most strongly supported model. Colors represent relative ΔBIC or ΔAIC values within clades. The data underlying this figure can be found in https://doi.org/10.5281/zenodo.14008084. **W Fig.** Variance explained ($R^2$) by clade-specific path models across tetrapods. Bars represent $R^2$ values for individual clades, reflecting the combined contribution of environmental, evolutionary time, and speciation rates predictors. Diamonds indicate mean $R^2$ values for each tetrapod class. The data underlying this figure can be found in https://doi.org/10.5281/zenodo.14008084. **X Fig.** Model fit statistics for clade-specific path models across amphibians. Distributions of goodness-of-fit indices for path models fitted separately to each clade, including the Comparative Fit Index (CFI), Root Mean Square Error of Approximation (RMSEA), and Standardized Root Mean Square Residual (SRMR). Each point represents the fit value for a given clade. Horizontal reference lines indicate commonly used thresholds for acceptable fit (CFI ≥ 0.95, RMSEA ≤ 0.05, SRMR ≤ 0.08). Most models meet or exceed recommended criteria for CFI and SRMR, whereas RMSEA values are frequently elevated, a pattern expected in large-sample and complex macroecological path models. The data underlying this figure can be found in https://doi.org/10.5281/ zenodo.14008084. **Y Fig.** Model fit statistics for clade-specific path models across reptiles. Distributions of goodness-of-fit indices for path models fitted separately to each clade, including the Comparative Fit Index (CFI), Root Mean Square Error of Approximation (RMSEA), and Standardized Root Mean Square Residual (SRMR). Each point represents the fit value for a given clade. Horizontal reference lines indicate commonly used thresholds for acceptable fit (CFI ≥ 0.95, RMSEA ≤ 0.05, SRMR ≤ 0.08). Most models meet or exceed recommended criteria for CFI and SRMR, whereas RMSEA values are frequently elevated, a pattern expected in large-sample and complex macroecological path models. The data underlying this figure can be found in https://doi.org/10.5281/zenodo.14008084. **Z Fig.** Model fit statistics for clade-specific path models across birds. Distributions of goodness-of-fit indices for path models fitted separately to each clade, including the Comparative Fit Index (CFI), Root Mean Square Error of Approximation (RMSEA), and Standardized Root Mean Square Residual (SRMR). Each point represents the fit value for a given clade. Horizontal reference lines indicate

commonly used thresholds for acceptable fit (CFI ≥ 0.95, RMSEA ≤ 0.05, SRMR ≤ 0.08). Most models meet or exceed recommended criteria for CFI and SRMR, whereas RMSEA values are frequently elevated, a pattern expected in large-sample and complex macroecological path models. The data underlying this figure can be found in https://doi.org/10.5281/zenodo.14008084. **AA Fig.** Model fit statistics for clade-specific path models across mammals. Distributions of goodness-of-fit indices for path models fitted separately to each clade, including the Comparative Fit Index (CFI), Root Mean Square Error of Approximation (RMSEA), and Standardized Root Mean Square Residual (SRMR). Each point represents the fit value for a given clade. Horizontal reference lines indicate commonly used thresholds for acceptable fit (CFI ≥ 0.95, RMSEA ≤ 0.05, SRMR ≤ 0.08). Most models meet or exceed recommended criteria for CFI and SRMR, whereas RMSEA values are frequently elevated, a pattern expected in large-sample and complex macroecological path models. The data underlying this figure can be found in https://doi.org/10.5281/zenodo.14008084. **AB Fig.** Relationship between clade age and the strength of direct predictor effects on species richness across tetrapods. Each point represents a clade, and lines show linear regressions between clade age and standardized effect sizes for environmental factors, evolutionary time, and speciation rates factors on species richness. The data underlying this figure can be found in https://doi.org/10.5281/zenodo.14008084. **AC Fig.** Relationship between clade physiology and the strength of direct predictor effects on species richness across tetrapods. Clades are classified as ectothermic or endothermic. Nonsignificant relationships are indicated by "ns". The data underlying this figure can be found in https://doi.org/10.5281/zenodo.14008084. **AD Fig.** Relationship between clade diversity (species richness) and the strength of direct predictor effects on species richness across tetrapods. Each point represents a clade, and lines show linear regressions between clade diversity and standardized effect sizes for environmental factors, evolutionary time, and speciation rates factors on species richness. The data underlying this figure can be found in https://doi.org/10.5281/zenodo.14008084. **AE Fig.** Relationship between clade geographic extent (range size) and the strength of direct predictor effects on species richness across tetrapods. Each point represents a clade, and lines show linear regressions between clade geographic extent and standardized effect sizes for environmental factors, evolutionary time, and speciation rates factors on species richness. The data underlying this figure can be found in https://doi.org/10.5281/zenodo.14008084. **AF Fig.** Robustness of path-model results to the number of Laplacian modalities (40 modalities). Bars show mean standardized path coefficients (β) representing direct, indirect, and total effects of five predictors on species richness across tetrapod clades identified using 40 modalities: amphibians (28 clades), reptiles (26 clades), birds (25 clades), and mammals (27 clades). Evolutionary time was approximated using mean pairwise phylogenetic distance (MPD) rather than assemblage arrival time, owing to the computational cost of historical biogeographic reconstructions. Other predictors include speciation rate (DR estimates), temperature, precipitation, and net primary productivity (NPP). Indirect effects are grouped into three classes: (i) environmental effects mediated through evolutionary time (MPD), (ii) environmental effects mediated through speciation rates, and (iii) climatic effects mediated through productivity. Total effects combine direct and all indirect pathways. This analysis evaluates whether reducing the number of modalities relative to the main analyses affects the overall conclusions. The data underlying this figure can be found in https://doi.org/10.5281/zenodo.14008084. **AG Fig.** Robustness of path-model results to the number of Laplacian modalities (60 modalities). Bars show mean standardized path coefficients (β) representing direct, indirect, and total effects of five predictors on species richness across tetrapod clades identified using 60 modalities: amphibians (35 clades), reptiles (34 clades), birds (34 clades), and mammals (37 clades). Evolutionary time was approximated using mean pairwise phylogenetic distance (MPD) rather than assemblage arrival time, owing to the computational cost of historical biogeographic reconstructions. Other predictors include speciation rate (DR estimates), temperature, precipitation, and net primary productivity (NPP). Total effects combine direct and indirect pathways. This analysis assesses whether increasing the number of modalities alters the qualitative conclusions. The data underlying this figure can be found in https://doi.org/10.5281/zenodo.14008084. **AH Fig.** Effects of environmental, evolutionary time, and speciation variables on species richness after excluding paraphyletic clades. Bars show mean standardized path coefficients (β) representing direct, indirect, and total effects across amphibians (30 clades), reptiles (31 clades), birds (29

clades), and mammals (25 clades). Predictors include evolutionary time (assemblage arrival time), speciation rate (DR estimates), temperature, precipitation, and net primary productivity (NPP). Indirect effects are grouped into three classes: (i) environmental effects mediated via evolutionary time, (ii) environmental effects mediated via speciation rates, and (iii) climatic effects mediated via productivity. Colours correspond to predictor variables included in the path models. The data underlying this figure can be found in https://doi.org/10.5281/zenodo.14008084. **AI Fig.** Direct effects of environmental, evolutionary time, and speciation variables on species richness estimated using molecular-only phylogenies. Clades were delineated using Laplacian spectral clustering applied to phylogenies restricted to species with genetic data only. Path models corresponding to Fig 3 were refitted for each clade, and mean standardized coefficients were extracted across clades for amphibians, reptiles, birds, and mammals. Points represent means and error bars indicate 95% confidence intervals. Colours denote effect direction and significance: blue, positive and significant; red, negative and significant; green, nonsignificant (confidence intervals overlapping zero). The data underlying this figure can be found in https://doi.org/10.5281/zenodo.14008084. **AJ Fig.** Indirect effects of climate (temperature and precipitation) on species richness mediated through productivity, estimated using molecular-only phylogenies. Clades were delineated using Laplacian spectral clustering applied to phylogenies restricted to genetically sampled species. For each clade, the path model shown in Fig 3 was fitted and standardized coefficients were extracted. Points represent mean effects across clades for each tetrapod class, with error bars indicating 95% confidence intervals. Colours denote effect direction and significance: blue, positive and significant; red, negative and significant; green, nonsignificant (confidence intervals overlapping zero). The data underlying this figure can be found in https://doi.org/10.5281/zenodo.14008084. **AK Fig.** Indirect effects of environmental variables mediated through evolutionary time (assemblage age) on species richness, estimated using molecular-only phylogenies. Clades were delineated using Laplacian spectral clustering applied to phylogenies restricted to genetically sampled species. For each clade, the path model shown in Fig 3 was fitted and standardized coefficients were extracted. Points represent mean effects across clades for each tetrapod class, with error bars indicating 95% confidence intervals. Colours denote effect direction and significance: blue, positive and significant; red, negative and significant; green, nonsignificant (confidence intervals overlapping zero). The data underlying this figure can be found in https://doi.org/10.5281/zenodo.14008084. **AL Fig.** Indirect effects of environmental variables mediated through speciation rates on species richness, estimated using molecular-only phylogenies. Clades were delineated using Laplacian spectral clustering applied to phylogenies restricted to genetically sampled species. For each clade, the path model shown in Fig 3 was fitted and standardized coefficients were extracted. Points represent mean effects across clades for each tetrapod class, with error bars indicating 95% confidence intervals. Colours denote effect direction and significance: blue, positive and significant; red, negative and significant; green, nonsignificant (confidence intervals overlapping zero). The data underlying this figure can be found in https://doi.org/10.5281/zenodo.14008084. **AM Fig.** Spearman correlations between assemblage age and phylometrics across amphibian clades. Points represent clade-specific correlations between assemblage age (defined as the mean arrival time of species within assemblages) and two phylogenetic metrics: mean pairwise phylogenetic distance (MPD) and maximum branch length (MBL). The dashed vertical line indicates zero correlation. Diamonds at the top of each panel show the mean correlation across clades for each metric, with horizontal error bars denoting 95% confidence intervals calculated using Fisher's *z*-transformation. The data underlying this figure can be found in https://doi.org/10.5281/zenodo.14008084. **AN Fig.** Spearman correlations between assemblage age and phylometrics across reptile clades. Points represent clade-specific correlations between assemblage age (defined as the mean arrival time of species within assemblages) and two phylogenetic metrics: mean pairwise phylogenetic distance (MPD) and maximum branch length (MBL). The dashed vertical line indicates zero correlation. Diamonds at the top of each panel show the mean correlation across clades for each metric, with horizontal error bars denoting 95% confidence intervals calculated using Fisher's *z*-transformation. The data underlying this figure can be found in https://doi.org/10.5281/zenodo.14008084. **AO Fig.** Spearman correlations between assemblage age and phylometrics across bird clades. Points represent clade-specific correlations between assemblage age (defined as the mean arrival time of species within

assemblages) and two phylogenetic metrics: mean pairwise phylogenetic distance (MPD) and maximum branch length (MBL). The dashed vertical line indicates zero correlation. Diamonds at the top of each panel show the mean correlation across clades for each metric, with horizontal error bars denoting 95% confidence intervals calculated using Fisher's $z$-transformation. The data underlying this figure can be found in https://doi.org/10.5281/zenodo.14008084. **AP Fig.** Spearman correlations between assemblage age and phylometrics across mammal clades. Points represent clade-specific correlations between assemblage age (defined as the mean arrival time of species within assemblages) and two phylogenetic metrics: mean pairwise phylogenetic distance (MPD) and maximum branch length (MBL). The dashed vertical line indicates zero correlation. Diamonds at the top of each panel show the mean correlation across clades for each metric, with horizontal error bars denoting 95% confidence intervals calculated using Fisher's $z$-transformation. The data underlying this figure can be found in https://doi.org/10.5281/zenodo.14008084. **AQ Fig.** Direct, indirect, and total effects of environmental, evolutionary time, and speciation rate on species richness using MPD as a proxy for evolutionary time. Bars show mean standardized path coefficients ($\beta$) across amphibian (34 clades), reptile (34 clades), bird (31 clades), and mammal (30 clades) clades. Evolutionary time is represented by mean pairwise phylogenetic distance (MPD), while additional predictors include speciation rate (DR estimates), temperature, precipitation, and net primary productivity (NPP). Indirect effects are grouped into three classes: environmental effects mediated via evolutionary time, environmental effects mediated via speciation rate, and climatic effects mediated via productivity. Total effects combine direct and all indirect pathways. Colours correspond to individual predictors. The data underlying this figure can be found in https://doi.org/10.5281/zenodo.14008084. **AR Fig.** Direct, indirect, and total effects of environmental, evolutionary time, and speciation rate on species richness using MBL as a proxy for evolutionary time. Bars show mean standardized path coefficients ($\beta$) across amphibian (34 clades), reptile (34 clades), bird (31 clades), and mammal (30 clades) clades. Evolutionary time is represented by maximum branch length (MBL), while additional predictors include speciation rate (DR estimates), temperature, precipitation, and net primary productivity (NPP). Indirect effects are grouped into three classes: environmental effects mediated via evolutionary time, environmental effects mediated via speciation rate, and climatic effects mediated via productivity. Total effects combine direct and all indirect pathways. Colours correspond to individual predictors. The data underlying this figure can be found in https://doi.org/10.5281/zenodo.14008084. **AS Fig.** Spearman correlations among speciation metrics across amphibian clades. Points represent clade-specific correlations between the DR statistic and alternative speciation rate estimates derived from BAMM or ClaDS. The dashed vertical line indicates zero correlation. Diamonds show mean correlations across clades with 95% confidence intervals derived using Fisher's $z$-transformation. Correlations were calculated across all grid cells occupied by each clade. The data underlying this figure can be found in https://doi.org/10.5281/zenodo.14008084. **AT Fig.** Spearman correlations among speciation metrics across reptile clades. Points represent clade-specific correlations between the DR statistic and alternative speciation rate estimates derived from BAMM or ClaDS. The dashed vertical line indicates zero correlation. Diamonds show mean correlations across clades with 95% confidence intervals derived using Fisher's $z$-transformation. Correlations were calculated across all grid cells occupied by each clade. The data underlying this figure can be found in https://doi.org/10.5281/zenodo.14008084. **AU Fig.** Spearman correlations among speciation metrics across bird clades. Points represent clade-specific correlations between the DR statistic and alternative speciation rate estimates derived from BAMM or ClaDS. The dashed vertical line indicates zero correlation. Diamonds show mean correlations across clades with 95% confidence intervals derived using Fisher's $z$-transformation. Correlations were calculated across all grid cells occupied by each clade. The data underlying this figure can be found in https://doi.org/10.5281/zenodo.14008084. **AV Fig.** Spearman correlations among speciation metrics across mammal clades. Points represent clade-specific correlations between the DR statistic and alternative speciation rate estimates derived from BAMM or ClaDS. The dashed vertical line indicates zero correlation. Diamonds show mean correlations across clades with 95% confidence intervals derived using Fisher's $z$-transformation. Correlations were calculated across all grid cells occupied by each clade. The data underlying this figure can be found in https://doi.org/10.5281/zenodo.14008084. **AW Fig.** Direct effects of environmental factors

(temperature, precipitation, and productivity), evolutionary time (assemblage age), and speciation rate on species richness after accounting for spatial structure. Path models include latitude and longitude as covariates to control for spatial autocorrelation. Points represent mean standardized path coefficients (β) across clades, with error bars indicating 95% confidence intervals, shown separately for amphibians, reptiles, birds, and mammals. Black diamonds indicate the mean effect size of each predictor across clades within each tetrapod class. Colours denote effect direction and statistical significance: blue, positive and significant; red, negative and significant; green, nonsignificant (confidence intervals overlapping zero). The data underlying this figure can be found in https://doi.org/10.5281/zenodo.14008084. **A Table.** Summary of taxonomic coverage for the four tetrapod classes included in the study. For each group, the table reports the total number of described species and the number of species retained in the analyses, defined by the intersection between available phylogenetic data and geographic range maps. **B Table.** Prior settings used in Bayesian Analyses of Macroevolutionary Mixtures (BAMM) for each tetrapod phylogeny. Priors were specified using the setBAMMpriors function in the BAMMtools package, with incomplete taxon sampling explicitly accounted for in all analyses.
(DOCX)

## Acknowledgments

We thank the members of the Storch lab for their insightful discussions on the manuscript.

## Author contributions

**Conceptualization:** Felipe Osmari Cerezer, David Storch.

**Data curation:** Felipe Osmari Cerezer, Antonin Machac, Jan Smyčka, Iñigo Rubio-López, Maxime Quétin.

**Formal analysis:** Felipe Osmari Cerezer, Antonin Machac, Jan Smyčka, Iñigo Rubio-López, Maxime Quétin.

**Investigation:** Felipe Osmari Cerezer, Antonin Machac, Jan Smyčka, Iñigo Rubio-López, Maxime Quétin.

**Methodology:** Felipe Osmari Cerezer, Jan Smyčka, David Storch.

**Supervision:** David Storch.

**Validation:** Felipe Osmari Cerezer, Jan Smyčka, Iñigo Rubio-López, Maxime Quétin, David Storch.

**Visualization:** Felipe Osmari Cerezer, Antonin Machac.

**Writing – original draft:** Felipe Osmari Cerezer, David Storch.

**Writing – review & editing:** Felipe Osmari Cerezer, Antonin Machac, Jan Smyčka, Iñigo Rubio-López, Maxime Quétin, David Storch.

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
