## [Editor Report · Decision Letter 0]

11 Sep 2025

Dear Dr Cerezer,

Thank you for submitting your manuscript entitled "Equilibrium dynamics shape diversity patterns across terrestrial tetrapod clades" for consideration as a Research Article by PLOS Biology.

Your manuscript has now been evaluated by the PLOS Biology editorial staff, as well as by an academic editor with relevant expertise, and I'm writing to let you know that we would like to send your submission out for external peer review.

Once your full submission is complete, your paper will undergo a series of checks in preparation for peer review. After your manuscript has passed the checks it will be sent out for review. To provide the metadata for your submission, please Login to Editorial Manager (https://www.editorialmanager.com/pbiology) within two working days, i.e. by Sep 15 2025 11:59PM.

Kind regards,

Roli Roberts

Roland Roberts, PhD

Senior Editor

PLOS Biology

rroberts@plos.org

---

## [Decision Letter · Decision Letter 1]

26 Nov 2025

Dear Dr Cerezer,

Thank you for your patience while your manuscript "Equilibrium dynamics shape diversity patterns across terrestrial tetrapod clades" was peer-reviewed at PLOS Biology. It has now been evaluated by the PLOS Biology editors, an Academic Editor with relevant expertise, and by four independent reviewers.

You'll see that reviewer #1 is positive about the overall study, but raises some significant concerns that need addressing; s/he questions the use of paraphyletic clades and worries about bias arising from imputation (and that the method is too sophisticated for the data quality). They suggest alternative methods, use of net diversification rates, ask about grid cell independence, and make a number of other suggestions. Reviewer #2 is very positive, with no substantive requests. Reviewer #3 is also broadly positive, but wonders about support for the claims and connection with empirical reality; their suggestions are constructive, but will need some further analysis and textual toning down. Reviewer #4 is extremely positive, but does worry about sources of bias and circularity, asking you to discuss this; the remaining concerns are textual and/or presentational.

In light of the reviews, which you will find at the end of this email, we would like to invite you to revise the work to thoroughly address the reviewers' reports.

Given the extent of revision needed, we cannot make a decision about publication until we have seen the revised manuscript and your response to the reviewers' comments. Your revised manuscript is likely to be sent for further evaluation by all or a subset of the reviewers.

**IMPORTANT - SUBMITTING YOUR REVISION**

*Re-submission Checklist*

*Published Peer Review*

*PLOS Data Policy*

*Blot and Gel Data Policy*

Sincerely,

Roland

Roland Roberts, PhD

Senior Editor

PLOS Biology

rroberts@plos.org

REVIEWERS' COMMENTS:

Reviewer #1:

I've read and reviewed the manuscript entitled "Equilibrium dynamics shape diversity patterns across terrestrial tetrapod clades" submitted as a research article for PLoS Biology. The manuscript presents a conceptual framework for testing the role of dynamic equilibria in governing species richness patterns in tetrapods. The paper's strengths lies in providing a thorough test of the role of environment, time, and rates, in a single conceptual framework, across a large number of clades. The paper is very well written and the figures are of the highest quality. The subject matter is of a broad interest to biologists and therefore is, in principle, suitable for publication in PLoS Biology. I do have quite a few large concerns about the data and analyses that I would hope to see addressed, I don't think any of these would be prohibitive to the authors, but greater consideration of what information is contained in a series of phylogenetic trees based on large amounts of imputed data, should be explicitly considered and accounted for. Overall, I congratulate the authors on a commendable effort here.

Main Comments

Clade Selection and Phylogenies

* I think the idea of including paraphyletic clades would make many people uneasy for this kind of analysis, and I don't think the argument pointing towards clades with extinct lineages really gets at the reason why. Lineages in the clade are clumped because of an abstract diversification modality rather than any biologically meaningful criteria (e.g., common traits or biogeography). What does it really mean to share a diversification modality and how distinct are they that they form distinguishable units? Would it be possible to just split each paraphyletic clade into a set of monophyletic clades, each sharing both a modality and common ancestry?

* Another possible issue with the Laplacian spectrum idea (which is very interesting by the way) is that many of the mega-trees used in this study contain a large number of species whose phylogenetic position are imputed. Therefore when producing an MCC tree from a posterior, you will find these species placed towards the "average" position of the taxonomic constraint they were imputed with - e.g., forming polytomies at the crown node of the genus or family (I know this because I've done it before). I worry that this will bias the spectrum as clades with high rates of imputation (and therefore high rates of polytomies / multiple species clustering around certain nodes) might fall within a similar modality. Or, the the modalities simply don't reflect true diversification scenarios.

* What this is really getting at, is that I think the method you've selected is too sophisticated for the data you have, which is largely imputed (especially for groups like amphibians). Best practice would be to find a way of clustering species into monophyletic clades based on some average modality membership across multiple phylogenies, or (more work) repeat downstream analyses across different trees from the posterior. Estimates of speciation rates and other metrics should be doing this in any case (i.e. not using MCC tree, but averaging values across posterior).

Speciation Rates

* The authors use speciation rates because they are more reliably estimated than extinction rates, citing Louca and Pennell etc. However, speciation rates are non-identifiable with extinction rates - meaning both rates can't be estimated from the infinite possible combinations that give the same net-diversification rate. This isn't a problem restricted to extinction rates. I would encourage the authors to use net diversification rates instead (this is what ClaDS and BAMM give you anyway for tip rates, I thought). I'm fine with the authors use the DR metric and stating it is known to be more strongly correlated with recent speciation rates in Title's study but they should use clear language regarding identifiability problems.

* Why is the grid cell the appropriate scale to test these hypotheses? They are not independent units of analysis because species contribute to multiple grid cells, and spatial variables are autocorrelated. How have the authors addressed these issues? Do the SEMs account for autocorrelation?

Clade-level traits

* Wouldn't geographic location / biome be more relevant that cool or warm epoch? There are cooler and warmer regions in any given epoch. This is also confounded with time because older origins = warmer, younger = cooler as seen in Figure 4. I don;t think this variable adds more information than time.

* Ancestral areas aren't really estimated because there is no information on the paleolocation of the areas themselves. This would be impossible to define in the first place because the areas aren't defined by anything that can be projected through time (e.g., climate), they are just contemporary representations of relatedness in space.

* While nice ideas in principle, I would consider removing the centroid displacement / rate / climate origin because I don't trust that they represent what you claim they represent.

* Further, linear models don't account for non-independence of clade's traits/geography/speciation rates - this should be tested with a pgls or similar model but paraphyletic clades mean this would be difficult to implement.

Path models

* Table 1 elegantly sets up four alternative models for the origin of richness. I would have expected the authors to fit these four alternative SEM's and use model selection to see which model provides the best fit, rather than just throwing every path in the same model which is far more difficult to interpret. Model selection would allow the authors to tally the number of clades which support each of the four scenarios.

* Path model statistics beyond the slopes were not presented in the paper. What about chi-square p value - do the models fit?

Minor Comments

* Why 50 clades? Was any test done to estimate optimal number of branching modes

* How many species were excluded by only retaining clades of 50 species or more - would this bias against recent + slow diversifying clades?

* Some recent criticisms about the meaning of grid-cell values of speciation rates are relevant here. Speciation happens at coarser scales than 1 degree grids.

* Paraphyletic clades again raise an issue with reconstructing historical biogeography, because tools like BGB assume monophyletic clades to estimate parameters

Figure 1.

* Label panels a/b/c/d or 1/2/3/4 (whichever is journal preference)

* Possibly remove extinction line from Historical Legacy and Diversity Cradles (or maybe indicate that it is not always diversity dependent - could make it a dashed line?)

Figure 2.

* Very nice figure!

Reviewer #2:

[identifies himself as Allen Hurlbert]

The authors unite four different hypotheses for diversity patterns within a single framework that considers the relative importance of equilibrial dynamics, speciation rate, and time. Through this unified framework, they are also able to evaluate the relative strength of direct and indirect paths of the relevant historical, evolutionary and environmental variables for a large number of terrestrial vertebrate clades. I would argue that this manuscript represents the best attempt to date to evaluate the relative importance of all of these hypotheses, and does so in a clear, elegant fashion. This paper is very well written, and the analyses and inferences provided here represent such an advance and clarification of a tangled literature that they are likely to end up in textbooks. Publish this.

Minor comments:

Section 2.7: In the context of traits, "size-related" can be interpreted as body size rather than clade size as you intend. Consider clarifying.

Reviewer #3:

Summary

The manuscript presents an innovative conceptual framework for distinguishing the mechanisms that generate species richness and applies it to a global analysis of tetrapods. Using a path modelling approach and clades defined by spectral partitioning, they simultaneously evaluate the role of evolutionary time, diversification rates, and environmental factors (productivity, temperature, precipitation). The results highlight a predominant role for productivity, exerting a dominant and direct effect on richness, while time and temperature play more contextual roles and speciation rates contribute little, reflecting mostly turnover dynamics. The work aims to reconcile classical explanations and provide an integrative framework for future studies.

General Comments

The manuscript is very well written and clearly reflects a large amount of work. However, in my view the authors make some very strong claims given the limitations of the available data. Much of the analysis relies on adjusting models (path equation model) to data created with their own inferences, on top of other models (e.g. phylogenies), which are then used again to interpret and explain the very patterns from which they were initially derived. This circularity makes it difficult to fully evaluate the robustness of the conclusions, and the underlying processes are not always clearly distinguished from the assumptions required to model them.

The study demonstrates impressive technical and computational capacity, but at times it feels somewhat disconnected from empirical grounding. For example, the biogeographical reconstructions are themselves estimates based on multiple assumptions, and the evolutionary variables are also estimates; building further interpretations on top of these layers of uncertainty leads to claims that may be overstated.

One methodological novelty is the selection of clades from phylogenies and the repeated analyses across clades to increase replication. This is indeed a strength of the approach, but it should be made clearer how this differs from previous work and how it affects the interpretation of the results.

Specific Comments

INTRODUCTION

-Add "1." before "Introduction", because the other paragraphs are numbered and the introduction is not (unify it).

-Unify the format of the references: remove the "," before the year in the following references: Yu & Wiens, 2024; Allen et al., 2006; Storch & Okie, 2019; Barreto et al., 2021.

-In the sentence "Although these explanations frequently operate together rather than as mutually exclusive alternatives (Cerezer et al. 2022; Machac 2020; Pontarp & Wiens 2017), their conceptual boundaries remain debated (Pontarp et al. 2019), highlighting the need to move beyond the traditional trichotomy." I suggest adding some classic references to provide a broader historical perspective, such as Pianka (1966, 1989), Fischer (1960), and Rosenzweig (1995).

MATERIALS AND METHODS

-In "We first delineated clades objectively using the Laplacian spectrum of large-scale phylogenies", I recommend removing the word "objectively", since the method itself is not objective but rather a mathematical procedure that summarizes tree properties and allows comparisons across trees. Please also add the citation for the method (Characterizing and Comparing Phylogenies from their Laplacian Spectrum).

Section 2.4. Estimating assemblage age:

-Please add in the Supplementary Material a map of this variable, as well as the evoregions from Nakamura et al., so readers can better visualize the approach.

-In the sentence "This approach explicitly estimates the timing and likely regional arrival of lineages into assemblages", I suggest replacing "explicitly estimates" with "infers" or "hypothesizes", since the method relies on multiple assumptions.

-In reporting temporal resolution, instead of "1 × 10⁻⁵ million years", please simplify and write "10 years". Avoid overcomplicating the text unnecessarily.

-While innovative, this metric is extremely complex and based on several assumptions, including reliance on a single phylogenetic hypothesis. Past ranges are inferred only from present distributions, which is a weak proxy for reconstructing historical dynamics or constraints. The variable is presented as if it were directly encoding "history," but in fact it does not represent any historical data. What it truly captures are present-day ranges combined with environmental variables (temperature, precipitation), whereas all historical aspects are hypotheses derived from phylogenies and ancestral reconstructions (i.e., models built on models, with partial data from the present). I recommend lowering the tone in this section, being more cautious, and recognizing that this variable is not a direct measure of history but rather a hypothesis about when species might have arrived in certain regions, strongly dependent on modeling assumptions. It seems premature to present it as if it were an empirical historical variable.

DISCUSSION

-I would perhaps change this sentence: "By contrast, evolutionary time and temperature influence richness more weakly and in a context-dependent manner." For: "In contrast, evolutionary time and temperature show weaker, context-dependent effects on richness."

CONCLUSIONS

-Add a "." at the end of the Conclusions paragraph (after the "and provide testable predictions to resolve long-standing debates in macroecology and macroevolution").

The study offers a useful integrative snapshot, but its inferences are inherently correlational: the "equilibrium" narrative is inferred from patterns of standardized path coefficients, not from explicit demographic/speciation-extinction dynamics.

Uncertain "assemblage age" hinges on DEC ancestral reconstructions and region definitions, while tip-based speciation metrics (DR/BAMM/ClaDS) emphasize recent rates and omit extinction—so diversification inference is incomplete.

Analyses on 1° grids might have inflated significance from spatial autocorrelation (also, please, specify in methods not in sup material that the maps have an equal area projection). Multicollinearity, measurement error in environmental layers, and residual correlations across equations can bias coefficients; reporting fit indices and comparing constrained SEMs that encode the four scenarios would strengthen causal claims.

Results from all clades are presented as averages, not including

Also, <50% of the variance is explained under their assumptions for all clades (36-46%), so, the title might be even the opposite. 50% of variance is not explained by productivity.

The fact that productivity is linked to richness is not a novel result.

Finally, clade selection via Laplacian spectra (including some paraphyly) may influence results and should be tested for sensitivity to alternative clade definitions.

Reviewer #4:

[identifies himself as Richard Field]

Signed review by Richard Field. (And I apologise for taking quite a long time to do this review.)

I think this is a wonderful paper! It addresses a classic question in ecology (what causes the latitudinal biodiversity gradients and related patterns? - often considered the 'holy grail of ecology') and provides both a framework for making progress on the theory and empirical results that address key hypotheses. Papers whose conceptual contribution is a mix of methodological and empirical are often difficult for journal editors because usually either, often both, of these advances is rather weak. This is a rare example of a paper in which each of these elements is, in my opinion, a sufficient advance *on its own* to merit publication in a very good journal like PLoS Biology. Putting the two together makes this one of the best and most important papers I have seen in a long time.

In addition, I congratulate the authors on having the manuscript in such good shape on first submission to the journal. I suppose that this is either a remarkable achievement by careful authors or the paper has previously been reviewed at one of the very top journals and improved as a result; it may be both. I would not be surprised if it had undergone revision for one of the very top journals because the conceptual advance provided is so significant. I can also envisage that rejection from such a journal might result because of some matter of reviewer taste (e.g. on very specific aspects of methods or not liking the implications), given how many areas of science this draws from and the implications that many peoples' preferred theories are shown rather convincingly herein to be modifiers rather than the main story. Whatever the pathway to submission to PLoS Biology, the result is that I have unusually little to criticise! (I normally provide many pages of pretty fundamental criticisms.)

I do have a few, relatively minor comments that the authors can consider. I did not find any line numbers in the manuscript, so below I include quotations, where appropriate, to aid location in the manuscript.

My most general comment is to wonder how fully the authors have thought through possible biases and the potential for circularity. Their review of limitations in the Discussion clearly demonstrates quite a lot of such thinking, and I find the arguments there convincing. From spending only a limited time (less than I would like) thinking about these while reading the manuscript, I am not convinced that there are any major bias or circularity problems. But I encourage the authors (who know their data and analyses much more thoroughly than I do) to consider whether any of the following aspects of their methods and data might lead to any cause for concern about bias or circularity - and either add some sentences to the Discussion accordingly, or if necessary do a few more sensitivity analyses. (I note that there are already lots of sensitivity analyses!)

* The range maps used. I think that they reflect a mix of observed occurrence data and then some form of interpolation - whether formal statistical interpolation or implicit interpolation by expert judgment, or both. Either interpolation method would typically use environmental variables either formally (e.g. by using elevation, precipitation, temperature, etc in an interpolation model) or informally (e.g. by an expert associating a species they know with shaded, wet habitats in mid elevations and mapping the range accordingly). Does the use of the range maps introduce any possible bias or circularity into the results? I'm not sure, but it might.

* "Historical influences were captured as the timing and likely regional arrival of lineages into assemblages (reconstructed via historical biogeographic models), while evolutionary effects were measured through tip-based speciation rates". I do not know the methods used here intimately enough to know whether any circularity or bias, with respect to the work done here, could creep in through them. I think probably not, but encourage the authors to think this through and respond.

SPECIFIC AND MINOR COMMENTS

I think it would be helpful to mention ecological opportunity somewhere in the Discussion and/or Introduction. The authors mention ecological saturation, which is kind-of the opposite, reducing speciation rates (or at least diversification rates). The other side of the coin is ecological opportunity promoting speciation rates, and this fits well with the authors' reasoning about the relationship with species richness, and the context dependency of the role of speciation rates. But, other than indirectly via a single citation of Schluter 2015, ecological opportunity is not actually mentioned in the manuscript.

On the subject of speciation rates, the authors say in the Intro: "Furthermore, recent studies challenge the assumption that speciation rates consistently predict richness, showing that they can be decoupled or even negatively correlated with species richness (Morlon 2020; Rabosky et al. 2018; Sun et al. 2020)." I do not think 'assumption' is the right word here - it is more about proposing this as an explanation than assuming it to be true in order to formulate some other explanation. I suggest the word 'notion' instead.

Table 1: It seems strange to me that the legends explaining the line colours are in the form of arrows coming out of Richness! Please modify.

Section 3.1: I like this opener to the Results section. It is a nice, clear statement uncluttered by detail. Even so I would refer to Fig.3 at the start of this paragraph

Section 3.2: in contrast, this section (and parts of 3.3) are highly cluttered with numbers, hindering readability. Figure 3 is much more effective at getting this information across to the reader (or it should be - see next point). I suggest removing the detailed numbers and instead focus on clearly and effectively narrating the key points in the text, letting Figure 3 take care of the fine details.

Figure 3 should show the confidence intervals (which are currently stated laboriously in the text). It is a basic principle of graphing that mean values should have appropriate error bars.

(Note here: the quite innovative approach that the authors used to generate the replication behind these mean values is to me a strength of the paper, giving the ability to apply meaningful confidence intervals - so it should come across in the graphical presentation of the results!)

Section 3.4 and the associated parts of the Discussion (e.g. start of 4.2). Yes there is a negative relationship between clade age and effect size, and the authors discuss it well. But I also see in Figure 4A a triangular (or constraining) relationship. There are no large effect sizes for old clades, while there is the full range of effect sizes for young clades. I think recognising this feature of the relationship and discussing it would enhance the manuscript. It is fully in line with the authors' interpretation of the results and actually I think would enhance that interpretation.

Figure 4 vs the text. Figure 4B has 'warming' and 'cooling' on the horizontal axis and the caption says "clades originating in warming vs. cooling periods". In contrast, throughout the text the wording used is 'warmer'/'cooler' and 'warm'/'cool'. Warm is not the same as warming, and cool is not the same as cooling - they are partially offset from each other in time. The authors should establish which it is and then align all the labels accordingly.

The first two sentences of the Discussion are Introduction material and repeat what has come before. I suggest deleting them and starting the discussion with "By applying a conceptual framework…"

While I agree with most, if not all of the interpretation and explanation in the paper, I am conscious that others may not. And it is important to stick to facts rather than interpretations when discussing things as facts. So the words 'strong influence of productivity' in the following sentence at the end of 4.1 concerns me:

"Overall, this perspective unifies three key observations: (i) the absence of a positive correlation between present-day speciation rates and species richness, (ii) the context-dependent effect of evolutionary time, and (iii) the consistently strong influence of productivity on richness patterns."

This sentence is defining key observations, but I do not agree that "the consistently strong influence of productivity on richness patterns" is an observation. It is an interpretation. The observation is the consistently strong positive relationship between productivity and richness patterns. So I suggest changing "consistently strong influence of productivity on richness patterns" to "consistently strong positive relationship between productivity and richness patterns".

On a similar note, the last paragraph of the Conclusion starts "Our analyses reveal that species richness across most tetrapod clades is primarily governed by equilibrium dynamics driven by productivity." Here I am concerned by the word 'reveal', for similar reasons as articulated above. How about changing the start of the sentence to "We infer from our analyses that species richness…"?

End of first paragraph of 4.2: "Figure S29-SX" needs fixing.

Start of second paragraph of 4.2: here I think it is worth mentioning and briefly discussing the exception, which is the moderately strong direct effect of precipitation on amphibian richness - in addition to the moderately strong indirect effect that all the groups have, which means that the total effect of precipitation is greater for amphibians than for the other groups. This makes ecological sense, given the particular requirements of most amphibians for water. Drawing some attention to this point not only fits with the existing discussion, but I think also would enhance the discussion in the paper.

Middle of last paragraph of Conclusion: "temperature exerts mainly direct effects, likely reflecting energetic constraints across trophic levels rather than indirect influences through productivity or diversification." When I read this it immediately seemed to me that it does not match how this is discussed in the Discussion, and checking back agrees with that. The last paragraph of 4.2 is the relevant part of the Discussion, and includes for example "A likely explanation is that temperature serves as a better proxy for resource availability than productivity (NPP) in many tetrapods". I suggest rewording the sentence in the Conclusion so that it matches what is in the Discussion.

Finally, I find the final sentence of the conclusion to be rather weak - vague and a bit of arm-waving. In contrast, the previous sentence is strong. I suggest deleting the final sentence, and finishing on the strength.

---

## [Decision Letter · Decision Letter 2]

27 Feb 2026

Dear Dr Cerezer,

Thank you for your patience while we considered your revised manuscript "Equilibrium dynamics governs diversity patterns in terrestrial tetrapod clades" for publication as a Research Article at PLOS Biology. This revised version of your manuscript has been evaluated by the PLOS Biology editors, the Academic Editor, and two of the original reviewers.

Based on the reviews and on our Academic Editor's assessment of your revision, we are likely to accept this manuscript for publication, provided you satisfactorily address the remaining points raised by the reviewers and the following data and other policy-related requests.

IMPORTANT - please attend to the following:

a) Please address the remaing very minor points from the reviewers; note that this includes a minor typo in the Title.

b) Please address my Data Policy requests below; specifically, we need you to supply the numerical values underlying Figs 3, 4, 5, 6A, S1-S49, either as a supplementary data file or as a permanent DOI’d deposition. I note that you already have an associated Zenodo deposition (https://doi.org/10.5281/zenodo.17020157). Please could you confirm whether the data and code in this deposition are sufficient to recreate the Figures?

c) Please cite the location of the data clearly in all relevant main and supplementary Figure legends, e.g. “The data and code needed to generate this Figure can be found in "https://zenodo.org/records/17020157"

d) Please make any custom code available, either as a supplementary file or as part of your data deposition.

e) Please include the URLs of your funders in the Financial Disclosure statement.

f) We note that you have a large number of references in the supplementary material; these citations will not receive recognition unless they are also in the main references. If necessary, please simply move your supplementary methods into the main paper - we have no page limitations.

We expect to receive your revised manuscript within two weeks.

*Published Peer Review History*

*Press*

Sincerely,

Roli Rolberts

Roland Roberts, PhD

Senior Editor

rroberts@plos.org

PLOS Biology

DATA POLICY:

Regardless of the method selected, please ensure that you provide the individual numerical values that underlie the summary data displayed in the following figure panels as they are essential for readers to assess your analysis and to reproduce it: Figs 3, 4, 5, 6A, S1-S49. NOTE: the numerical data provided should include all replicates AND the way in which the plotted mean and errors were derived (it should not present only the mean/average values).

CODE POLICY

Per journal policy, if you have generated any custom code during the course of this investigation, please make it available without restrictions. Please ensure that the code is sufficiently well documented and reusable, and that your Data Statement in the Editorial Manager submission system accurately describes where your code can be found. More information on our Code Policy, what and how to share can be found here: https://journals.plos.org/plosbiology/s/code-availability

DATA NOT SHOWN?

REVIEWERS' COMMENTS:

Reviewer #1:

I appreciate the hard work the authors have put into these revisions. I was one of the reviewers who came on board at R1, and I know it can be annoying to get a new set of comments at that stage. I think this paper is very well written and exceptionally thorough. The authors ability to conduct a large number of robustness checks (removing paraphyletic clades, using molecular-only species in the phylogeny, choice of evolutionary-time metric, speciation rate metric, and effect of spatial autocorrelation) requested by myself and others gives me a lot of additional confidence in the results. This is going above-and-beyond and the authors deserve a lot of credit for this. For example, repeating the analysis on the molecular-only data gives me a lot more confidence in the Laplacian spectrum approach and softens my anxiety that the megaphylogenies used in this study might not be the best datasets for the question.

I'm also glad the authors liked the suggestion of the model selection for the SEM analysis. I feel like this does nicely streamline that section and make it easier to interpret as a reader.

I don't think I agree with the authors that non-identifiability propagates to net-diversification rates in the same way as for speciation and extinction rates, but I do think the additional text the authors have contributed is sufficient here, and the authors have done more than enough to satisfy any concerns I had.

Reviewer #4:

Signed review by Richard Field

I have read through all the previous reviews and the authors' responses to them, as well as the revised manuscript. As well as doing everything I asked of them, in my opinion, the authors have replied very thoroughly to all the comments in the reviews, taking all of them seriously, making appropriate revisions to the paper and doing a lot of new sensitivity analyses even though it seemed clear to me that the results would not change meaningfully - as has turned out to be the case.

In my view, this is ready to publish, pending a good copy edit, including correction of a typo in the title: change 'governs' to 'govern'.

---

## [Editor Report · Decision Letter 3]

14 Mar 2026

Dear Dr Cerezer,

Thank you for the submission of your revised Research Article "Diversity patterns in terrestrial tetrapod clades are governed by equilibrium dynamics" for publication in PLOS Biology. On behalf of my colleagues and the Academic Editor, Tiago Quental, I'm pleased to say that we can in principle accept your manuscript for publication, provided you address any remaining formatting and reporting issues. These will be detailed in an email you should receive within 2-3 business days from our colleagues in the journal operations team; no action is required from you until then. Please note that we will not be able to formally accept your manuscript and schedule it for publication until you have completed any requested changes.

Sincerely,

Roli Roberts

Senior Editor

PLOS Biology

rroberts@plos.org